# scDEBART: Predicting *in silico* Single-Cell Perturbation Responses via Large-Scale Differential Expression Learning

**Jieun Sung, Wankyu Kim** [* 1]

## Abstract

Single-cell foundation models trained on millions of cells can learn gene expression patterns across diverse contexts. However, for predicting genetic perturbation effects they often underperform simple regression models. We hypothesize two potential limitations: targets defined on dropout-prone absolute expression, and pretraining objectives that reconstruct static co-expression rather than encoding how genes co-regulate under expression changes. We introduce **scDEBART**, a perturbation-specific pretraining framework that predicts log fold-changes (logFC) conditioned on basal expression, thereby learning how gene sets co-vary across expression-change contexts at scale. To obtain reliable estimates of expression change under technical sparsity, we compute logFC from scVI-denoised expression and restrict pretraining to genes with robust detection. Pretrained on 6.28 million expression-change profiles from 66.6 million human cells and fine-tuned on five Perturb-seq datasets, scDEBART achieves mean enrichment factor (EF) of 11.96, 4–7$\times$ higher than scGPT and GEARS (mean EF 1.74–2.99), and 71.4% top-1 accuracy for reverse perturbation identification compared to near-zero accuracy for prior models. In cross-modal transfer to drug perturbations (SCIPLEX), the model shows dose-dependent improvement in directional alignment (cosine similarity 0.04$\rightarrow$0.30) with above-random DEG enrichment (EF 2.91–4.32), suggesting partial transfer of learned regulatory patterns across modalities. Overall, these results indicate that large-scale pretraining on scVI-denoised expression-change profiles provides a useful inductive bias for perturbation prediction.

[1]Department of Life Science, Ewha Womans University, Seoul, Republic of Korea. Correspondence to: Wankyu Kim <wkimwkim@gmail.com>.

*Proceedings of the 43$^{rd}$ International Conference on Machine Learning*, Seoul, South Korea. PMLR 306, 2026. Copyright 2026 by the author(s).

## 1. Introduction

Large foundation models have transformed natural language processing and are now being explored in single-cell transcriptomics: models such as GeneFormer (Theodoris et al., 2023) and scGPT (Cui et al., 2024) pretrain on millions of single-cell RNA-sequencing (scRNA-seq) profiles to learn gene expression patterns across cells. Self-attention enables these models to capture gene–gene dependencies, yielding embeddings that summarize transcriptional state and functional context, which can be fine-tuned for tasks including cell type classification, batch correction, and perturbation-effect prediction.

A particularly important application is predicting transcriptional responses to genetic perturbations (*in silico* knockout or overexpression). Typical approaches modify input expression (e.g., zeroing a gene for knockout or increasing it for overexpression) and then forecast transcriptome-wide changes, which can aid in inferring gene function and prioritizing therapeutic targets. However, recent work has shown that current single-cell foundation models often fail to outperform simple regression models on this task (Ahlmann-Eltze et al., 2025), motivating a closer examination of the design choices that may underlie this gap.

We hypothesize this gap stems partly from two design choices in existing approaches. First, models predict absolute expression values and compute differential expression post hoc by comparing predicted perturbed and unperturbed states. Because scRNA-seq data exhibits severe dropout and zero inflation (Qiu, 2020), fold-changes derived from point predictions can be unstable, as technical zeros may dominate the signal and distort inferred perturbation-induced changes. Second, models are typically pretrained to reconstruct masked gene expression within individual cells, learning static co-expression patterns. While these patterns capture which genes tend to be co-expressed, they may not directly encode how genes co-regulate in response to expression changes—the regulatory dynamics relevant for perturbation prediction (Li et al., 2023; Dixit et al., 2016; Replogle et al., 2022).

These observations led us to ask whether reframing the pretraining objective—from reconstructing absolute expres-

sion within cells to predicting expression changes across cell populations—could provide a more suitable inductive bias for perturbation prediction. Here, we propose **scDE-BART** (single-cell Differential Expression Bidirectional AutoRegressive Transformer), a perturbation-specific pretraining framework that predicts population-level log fold-changes (logFC) conditioned on basal expression. We posit that learning how gene sets co-vary across diverse expression contexts provides a more suitable inductive bias for perturbation prediction than reconstructing static expression snapshots. While these naturally occurring differences are observational rather than causal interventions, we hypothesize that genes with consistent co-variation patterns across diverse basal contexts encode regulatory principles relevant to perturbation responses. By pretraining to reconstruct logFC conditioned on basal expression across millions of such comparisons, scDEBART learns which genes respond together under diverse contexts, capturing regulatory co-variation closer to what perturbations elicit.

To make logFC supervision robust under technical sparsity, we compute logFC from scVI-denoised expression (Lopez et al., 2018) rather than dropout-prone raw counts, and restrict training to genes with reliable detection (non-zero proportion $\geq 0.3$) and high differential expression confidence (Proba-DE $\geq 0.8$, $|\text{FC}| \geq 1.5$), reducing the influence of technical zeros. Using this pipeline, we assemble 6.28 million differential expression profiles from 66.6 million cells across 617 datasets for pretraining. We then fine-tune scDEBART on multiple Perturb-seq datasets—Norman K562 (Norman et al., 2019), Replogle K562 and RPE1 (Replogle et al., 2022), and Nadig HepG2 and Jurkat (Nadig et al., 2025)—and evaluate transfer to drug perturbations in SCIPLEX (Srivatsan et al., 2020) to probe generalization beyond CRISPR-based perturbations.

Our key contributions are:

- **Perturbation-specific pretraining on expression changes at scale.** We pretrain a BART transformer to predict logFC conditioned on basal expression, using 6.28 million differential expression profiles derived from 66.6 million human cells across 617 datasets. Unlike prior single-cell foundation models that reconstruct static expression, this objective is explicitly designed to capture co-regulatory gene dynamics aligned with the perturbation prediction task.

- **Robust differential expression supervision.** We compute logFC from scVI-denoised expression rather than dropout-prone raw counts, and restrict pretraining to genes with reliable detection and high differential expression confidence, mitigating the influence of technical zeros.

- **Superior perturbation prediction and cross-modal**

**transfer.** On five Perturb-seq benchmarks, scDEBART achieves 4–7$\times$ higher enrichment for DEG recovery than scGPT and GEARS (mean EF 11.96 vs. 1.74, 2.99 at top-50 genes) and 71.4% top-1 accuracy for reverse perturbation identification compared to near-zero accuracy for prior models. In cross-modal transfer to drug perturbations (SCIPLEX), a model trained only on genetic perturbations achieves higher enrichment than baselines, suggesting partial transfer of learned regulatory patterns across perturbation modalities.

## 2. Related Works

### 2.1. Denoising and Differential Expression for scRNAseq data with scVI

Lopez et al. introduced scVI (single-cell Variational Inference) (Lopez et al., 2018), a variational autoencoder (VAE)-based generative model that separates biological variation from technical confounders in scRNA-seq data. scVI models gene counts with a negative binomial family likelihood and learns cell-specific latent representations together with gene-specific parameters such as mean expression and dispersion, while accounting for technical factors including library size and batch effects via covariate conditioning. For differential expression (DE) analysis, scVI compares cell populations through posterior inference rather than relying solely on point estimates, enabling uncertainty-aware estimates of log fold-change and related statistics. This approach can yield more stable DE estimates under technical sparsity, especially when raw counts are dominated by zeros.

### 2.2. Previous Models for Perturbation Prediction

**GEARS** Roohani et al. proposed GEARS (Graph-Enhanced gene Activation and Repression Simulator) (Roohani et al., 2024) for predicting the effects of single- and multi-gene perturbations. GEARS integrates a prior gene–gene interaction graph with graph neural networks (GNNs), where nodes represent genes and edges encode known regulatory relationships. By combining gene embeddings with perturbation-specific representations through message passing on the interaction graph, GEARS aims to generalize to unseen perturbation combinations via compositional structure.

**scGPT** Cui et al. developed scGPT (Cui et al., 2024), a transformer-based foundation model pretrained on large-scale scRNA-seq corpora. Treating genes as tokens, scGPT learns contextual representations that capture gene dependencies across diverse tissues and cell types, and has been adapted to a range of downstream tasks including cell type annotation, integration, and perturbation-response prediction.

While these models provide powerful representations of tran-

scriptional state, recent benchmarks suggest that they do not consistently outperform simpler baselines on perturbation-effect prediction (Ahlmann-Eltze et al., 2025), indicating room for methodological improvement.

## 3. Methods

### 3.1. Construction of Differential Expression Corpus

#### 3.1.1. DATA COLLECTION AND PREPROCESSING

To construct a comprehensive corpus of differential expression (DE) profiles, we curated publicly available scRNA-seq datasets from the CELLxGENE data portal. As of January 30, 2025, the repository contained 1,573 datasets (893 human) with over 106 million cells. We applied dataset-level filtering criteria: (i) inclusion of normal (healthy) human cells to reduce learning disease-specific artifacts; and (ii) dataset size between 10,000 and 1,000,000 cells to balance computational cost and statistical stability.

This filtering yielded 617 datasets comprising 66,611,859 cells across 61 tissue types, 81 disease types, and 23 sequencing technologies. For each dataset, we retained only protein-coding genes and applied adaptive cell-level quality control based on total UMI counts and mitochondrial read fraction, keeping cells within $\pm 3$ standard deviations of dataset-specific means as in (Theodoris et al., 2023).

#### 3.1.2. CLUSTERING CELLS AND DE ANALYSIS

**Clustering cells and selecting cluster pairs** After quality control, we employed SECUER (Wei et al., 2022), a density-based clustering algorithm, to identify transcriptionally coherent cell populations, yielding 98,753 clusters across all datasets with a median size of 278 cells.

For each dataset, we trained an scVI model (Lopez et al., 2018) with 64 latent dimensions to account for technical factors and extract latent representations. To select informative cluster pairs, we computed cluster centroids as mean latent vectors of cells within each cluster and restricted pairwise cosine similarities to the 10th–90th percentile range. This heuristic avoids near-identical pairs with negligible signal and extremely dissimilar pairs that likely reflect broad, multifactor differences rather than localized regulatory shifts.

**DE profile construction and filtering.** For each selected cluster pair $(A, B)$, we performed scVI posterior-sampling differential expression analysis and computed per-gene statistics including (i) median log fold-change $x_i$ (logFC), (ii) scVI-denoised expression $\mu_i$ of the baseline cluster, (iii) differential expression probability (Proba-DE), and (iv) detection rate (non-zero proportion). We define a *DE profile* as an ordered list of genes ranked by $|x_i|$ and paired with these statistics. To mitigate the influence of unreliable esti-

mates, we retained only genes satisfying: Proba-DE $\geq 0.8$, non-zero proportion $\geq 0.3$ in both clusters, and $|FC| \geq 1.5$. Profiles with fewer than 10 retained genes were discarded. Each profile was truncated to at most $L = 1{,}024$ genes, yielding 6,282,796 DE profiles covering 17,944 genes. Detailed corpus construction, clustering, scVI training, cluster-pair selection, and filtering rationale are provided in Appendix A.1.

### 3.2. Pretraining scDEBART

scDEBART is designed to model expression changes conditioned on basal expression states using a BART encoder–decoder architecture (Lewis et al., 2019). Instead of predicting absolute expression and deriving fold-changes post hoc, the model is pretrained to reconstruct masked logFC values from gene priors and basal expression, directly aligning the pretraining objective with perturbation prediction.

**Gene representation.** To incorporate biological knowledge, we constructed gene-level representations from BioBERT functional embeddings (768-dim) and MSigDB pathway membership (7,411-dim) as biological priors (Appendix A.3.1).

Each gene $g_i$ in a DE profile of length $L \leq 1024$ is represented by fusing these biological priors with scVI-derived features. Let $d = 768$ denote the model dimension. We construct

$$\mathbf{h}_i = W_p \left[ W_k \left( [W_r \mathbf{r}_i \,;\, \mathbf{b}_i] \right) \,;\, W_{\text{logFC}} x_i \,;\, W_{\text{expr}} \log(1 + \mu_i) \right],$$

where $\mathbf{b}_i \in \mathbb{R}^{768}$ is the BioBERT embedding, $\mathbf{r}_i$ is the MSigDB pathway membership vector, $x_i$ is logFC, $\mu_i$ is baseline normalized expression, and $W_r, W_k, W_{\text{logFC}}, W_{\text{expr}}, W_p$ are learnable projections.

**Encoder and decoder.** The encoder has $N_{\text{enc}} = 8$ Transformer layers (12 heads, head dimension 64) with bidirectional self-attention, and the decoder has $N_{\text{dec}} = 4$ layers with masked self-attention and encoder–decoder cross-attention. Two MLP heads operate sequentially on decoder outputs: (i) an embedding reconstruction head $(d \rightarrow 2d \rightarrow d)$ that produces $\hat{\mathbf{h}}_i$ from the decoder output $\mathbf{d}_i$, and (ii) a logFC prediction head $(d \rightarrow 512 \rightarrow 1)$ that takes $\hat{\mathbf{h}}_i$ as input to predict $\hat{x}_i$, both with GELU activations.

**Pretraining objective and masking.** We pretrain scDEBART on the 6.28M-profile DE corpus using a denoising objective. For each profile, we randomly select a masked set $\mathcal{M}$ comprising 15% of genes (excluding padding positions). We apply masking by setting the input logFC value to zero and overwriting the corresponding logFC embedding with an all-zero vector at masked positions, while keeping gene priors and baseline expression unchanged. The model is

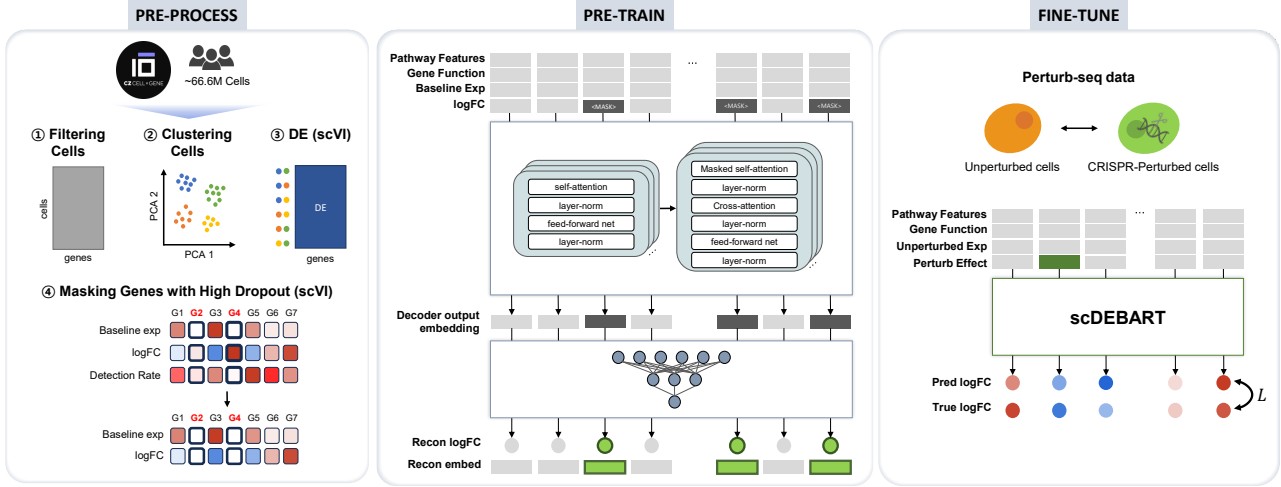

Figure 1. **scDEBART workflow.** We construct a corpus of 6.28 million differential expression (DE) profiles from 66.6 million cells in CELLxGENE through quality filtering, cell clustering, scVI-based DE analysis, and stringent gene filtering. A BART encoder–decoder is pretrained to reconstruct masked logFC values conditioned on gene priors and basal expression, then fine-tuned on Perturb-seq data to predict perturbation-induced transcriptome-wide responses.

then trained to reconstruct logFC and embedding only for the masked genes, using the unmasked genes as context.

The model is trained with a combined reconstruction loss:

$$\mathcal{L} = \alpha \cdot \mathcal{L}_{\text{logFC}} + (1 - \alpha) \cdot \mathcal{L}_{\text{emb}},$$

where $\mathcal{L}_{\text{logFC}} = \frac{1}{|\mathcal{M}|} \sum_{i \in \mathcal{M}} (x_i - \hat{x}_i)^2$ and $\mathcal{L}_{\text{emb}} = \frac{1}{|\mathcal{M}|} \sum_{i \in \mathcal{M}} \|\mathbf{h}_i - \hat{\mathbf{h}}_i\|_2^2$, with $\alpha = 0.8$.

Full hyperparameters and training details are in Appendix A.3.

### 3.3. Fine-tuning on Perturb-seq Datasets

**Dataset preparation and filtering.** We fine-tuned scDE-BART on five CRISPR-based Perturb-seq datasets: Norman K562 (Norman et al., 2019), Replogle K562 and RPE1 (Replogle et al., 2022), and Nadig HepG2 and Jurkat (Nadig et al., 2025). Norman K562 applied CRISPRa to overexpress 284 perturbations (single and double) in K562 chronic myeloid leukemia cells. The Replogle datasets (K562 and RPE1) and Nadig datasets (HepG2 and Jurkat) used genome-scale CRISPRi to knock down approximately 1,500–2,600 genes across K562, RPE1 (retinal pigment epithelial), HepG2 (hepatocellular carcinoma), and Jurkat (T-lymphocyte) cell lines. First, we applied quality control procedures identical to the CELLxGENE corpus, then retained only cells where target gene expression was effectively perturbed and predicted to be high CRISPR efficacy using the Perturbation Score (PS) (Song et al., 2025) (Appendix A.2.1). To further reduce contamination from weakly perturbed cells, we additionally kept cells belonging to the major response-consistent cluster in latent space. This filtering yielded between 40,500 and 157,676 high-quality cells per dataset (Appendix Table 1). Details are in Appendix A.2.

**Fine-tuning input and objective.** For a perturbation targeting gene $g_{\text{pert}}$, the model predicts a logFC vector $\hat{\mathbf{x}} = [\hat{x}_{g_1}, \ldots, \hat{x}_{g_N}]$ over the selected gene set (we use up to $N = 5,000$ HVGs per dataset), with ground-truth logFC denoted as $\mathbf{x}^* = [x_{g_1}^*, \ldots, x_{g_N}^*]$. For each gene $g_i$, we replace the logFC input with a perturbation indicator $p_i$:

$$\mathbf{h}_i^{\text{ft}} = W_p \left[ W_k \left( [W_r \mathbf{r}_i \, ; \, \mathbf{b}_i] \right) \, ; \, W_{\text{logFC}} p_i \, ; \, W_{\text{expr}} \log(1 + \mu_i) \right],$$

where $p_i$ is set to $p_{\text{inh}} = -10.0$ for inhibition and $p_{\text{act}} = +10.0$ for activation if $g_i = g_{\text{pert}}$, and $p_i = 0$ otherwise. Using large-magnitude constants provides a clear conditioning signal well-separated from typical pretraining logFC values, helping the model distinguish fine-tuning perturbation indicators from observed logFC inputs.

Unlike pretraining, which uses a masked denoising objective, fine-tuning optimizes a composite loss combining weighted MSE and correlation objectives to balance magnitude accuracy and ranking performance:

$$\mathcal{L}_{\text{finetune}} = \alpha \cdot \mathcal{L}_{\text{wMSE}} + \gamma \cdot \mathcal{L}_{\text{corr}},$$

where the weighted MSE loss places higher weight on genes with larger fold-changes:

$$\mathcal{L}_{\text{wMSE}} = \frac{1}{N} \sum_{i=1}^{N} \left( 1 + \lambda \cdot |x_{g_i}^*| \right) \cdot (\hat{x}_{g_i} - x_{g_i}^*)^2, \quad \lambda = 0.5,$$

and the correlation loss combines global and top-K Pearson correlation:

$$\mathcal{L}_{\mathrm{corr}} = 1 - [\beta \cdot \rho_{\mathrm{global}}(\hat{\mathbf{x}}, \mathbf{x}^*) + (1 - \beta) \cdot \rho_{\mathrm{top}}(\hat{\mathbf{x}}, \mathbf{x}^*)],$$

where $\rho_{\mathrm{global}}$ is computed over all genes and $\rho_{\mathrm{top}}$ is computed on the top-20% genes ranked by $|\mathbf{x}^*|$. We treat $\alpha$, $\beta$, and $\gamma$ as learnable scalar weights (parameterized via sigmoid transforms and initialized to 1.0, 0.5, and 0.5, respectively) and jointly optimize them with the model parameters. We split each dataset 80/10/10 (train/val/test) at the gene level (Roohani et al., 2024) and fine-tuned using AdamW (learning rate $10^{-4}$) with mixed precision. All experiments were performed with three random seeds to ensure robustness. Details are in Appendix A.4.

### 3.4. Cross-Modal Transfer on Drug Perturbations

To assess whether gene regulatory relationships learned from CRISPR-based perturbations transfer to drug-induced responses, we performed cross-modal transfer on SCI-PLEX (Srivatsan et al., 2020), a large-scale drug perturbation dataset comprising 188 compounds across three cell lines (K562, A549, MCF7) at four concentrations (10, 100, 1000, 10000 nM) with 24-hour perturbation times. We retained only K562 cells to match the cell line used in Replogle K562 fine-tuning, yielding 158,293 cells after quality control (cells with total UMI counts or mitochondrial read fractions beyond ±3 standard deviations were removed). Genes were filtered to protein-coding genes in the scDE-BART vocabulary (19,742 genes).

For each compound, we mapped to primary protein targets using DrugBank (Knox et al., 2024), retaining only targets annotated as "inhibitor" or "antagonist" to align with CRISPRi knockdown mechanisms. We simulated multi-gene perturbation by setting perturbation indicators $p_i = -10.0$ for all target genes and $p_i = 0$ otherwise, matching the Replogle K562 fine-tuning protocol. For compounds with multiple targets, all targets were perturbed simultaneously as a multi-gene knockout.

We computed drug-induced logFC from SCIPLEX using scVI-based differential expression analysis, following the same protocol as Perturb-seq datasets (Appendix A.5.2). We evaluated predicted responses using the same metrics as genetic perturbation evaluation (cosine similarity, RMSE, enrichment factor, F1 score) on top-20 DEGs across all four doses. Detailed target mapping and evaluation protocols are in Appendix A.5.

## 4. Results

### 4.1. Perturbation Response Prediction

scDEBART was pretrained on 6.28M DE profiles from 66.6M cells (Figure 1), using scVI-denoised logFC with stringent filtering (non-zero proportion $\geq 0.3$; Proba-DE $\geq 0.8$, $|\mathrm{FC}| \geq 1.5$; see Methods). The model learns to reconstruct masked logFC values conditioned on basal expression and gene priors, capturing how gene sets co-vary across diverse cellular contexts while mitigating dropout artifacts through corpus-level filtering.

To evaluate prediction accuracy on genetic perturbations, we assessed performance on five large-scale Perturb-seq datasets (Norman et al., 2019; Replogle et al., 2022; Nadig et al., 2025) (CRISPRi knockdowns and CRISPRa over-expression). We compared scDEBART against GEARS, scGPT, multi-layer perceptron (MLP), and linear regression (LR) baselines under two preprocessing regimes: scVI-denoised expression (scVI-[model]) and library-size normalized counts (counts-[model]). This design disentangles performance gains from preprocessing (scVI denoising) versus model architecture. We evaluated held-out test genes (80/10/10 splits) using four metrics on top-$K$ genes ($K \in \{50, 100, 200, 300, 400, 500\}$): cosine similarity and RMSE for directional agreement and magnitude accuracy, and enrichment factor (EF) and F1 for DEG recovery and directional classification (1.2-fold threshold; Appendix A.4.3). All metrics were computed on logFC, which more directly reflects perturbation effects (Wei et al., 2025); baseline models that predict absolute expression were therefore converted to logFC via matched controls (Appendix A.4.4).

**scDEBART demonstrates strong performance across metrics.** Figure 2 shows that scDEBART outperformed all baselines on enrichment factor, RMSE and F1 score across all five datasets. For cosine similarity at top-50 genes, scDEBART ranked first on two datasets (Norman K562: 0.777 vs. second-place counts-LR 0.721, +7.8%; Replogle K562: 0.737 vs. scVI-scGPT 0.685, +7.6%) and second on three datasets (Replogle RPE1: 0.801 vs. scVI-scGPT 0.842, −4.9%; Nadig HepG2: 0.787 vs. scVI-scGPT 0.803, −2.0%; Nadig Jurkat: 0.660 vs. scVI-scGPT 0.682, −3.2%). The most substantial gains appeared in enrichment factor: scDEBART achieved EF of 7.73–16.14 compared to scVI-GEARS (1.67–4.29), scVI-scGPT (0.69–3.36), and counts-based variants (0.55–1.12), exceeding the best-performing baseline by 2.3–26.3× on each dataset. RMSE was consistently lower for scDEBART (0.53–1.01) than for all baselines, with the next-best baseline achieving 0.70–1.06. F1 scores ranged from 0.35–0.61 for scDEBART, higher than all baselines (0.03–0.36).

**Preprocessing effects depend on pretraining status.** Among count-based models, simpler architectures (MLP, LR) outperformed complex models (GEARS, scGPT), with mean top-50 cosine similarity of 0.63 and 0.62 for counts-MLP and counts-LR respectively, versus 0.14 and 0.45 for counts-GEARS and counts-scGPT, consistent with (Ahlmann-Eltze et al., 2025). This pattern suggests that

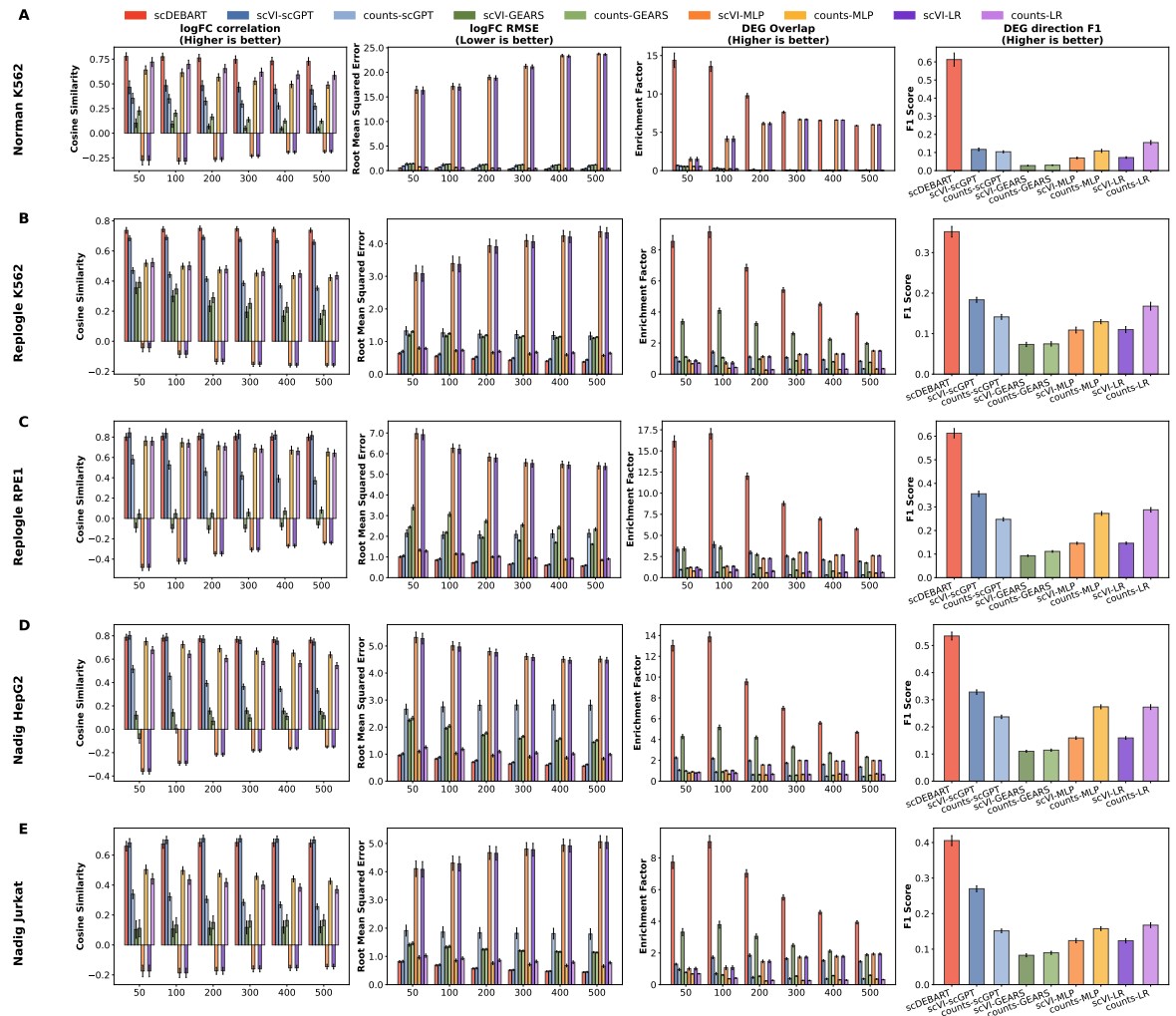

*Figure 2.* **Perturbation response prediction across five Perturb-seq datasets.** scDEBART is compared with eight baselines (GEARS/scGPT/MLP/LR, each evaluated with scVI-denoised expression and library-size normalized count). We report cosine similarity, RMSE, enrichment factor, and F1 score evaluated on top-$K$ DEGs (see Appendix A.4.3 for details). Bars indicate mean performance over all test perturbations across three random seeds, with error bars showing 95% confidence intervals.

under dropout-prone raw counts, low-capacity models act as more empirically robust regressors. However, scVI denoising had opposite effects depending on pretraining status. The pretrained scGPT benefited substantially from denoising (scVI-scGPT 0.70 vs. counts-scGPT 0.45, a 54% relative increase), suggesting that denoising may allow pretrained gene dependency patterns to function more effectively by reducing dropout artifacts. In contrast, non-pretrained models showed either minimal change (scVI-GEARS 0.12 vs. counts-GEARS 0.14) or substantially degraded performance under denoising (scVI-MLP $-0.27$ vs. counts-MLP 0.63; scVI-LR $-0.27$ vs. counts-LR 0.62), indicating that denoising alone cannot overcome the lack of pretraining signal. This pattern indicates that denoising benefits pretrained models by aligning fine-tuning data with the distribution encountered during large-scale pretraining. In contrast, models without such priors struggle to extract regulatory patterns from denoised expression alone.

**Baseline models prioritize low-detection genes, producing false positives.** Despite competitive cosine similarity, baseline models showed markedly lower enrichment factors, suggesting they capture broad directional trends but fail to rank true DEGs highly. We examined detection rates of top-50 predicted genes (Appendix Figure 13): scDEBART selected genes with median non-zero proportions of 0.60 (perturbed) and 0.73 (control), whereas most baselines prioritized genes with substantially lower detection (0.09–0.28). This pattern persisted across both preprocessing regimes, indicating that scVI denoising alone cannot fully correct dropout-induced bias. scDEBART's explicit filtering during pretraining—restricting the training corpus to high-detection genes—shifted predictions toward reliably detected DEGs,

whereas most baselines continued to prioritize low-detection genes. Together, these results suggest that corpus-level filtering during pretraining is critical for preventing models from learning spurious correlations driven by dropout-induced noise. This bias toward low-detection genes in baseline models may reflect their inability to distinguish true biological effects from zero-inflation artifacts.

**scDEBART's predictions generalize across DE pipelines.** Evaluation using DESeq2-based pseudobulk differential expression as ground truth corroborated scDEBART's enrichment factor advantage across most datasets. For cosine similarity, scDEBART showed dataset-dependent performance, ranking first on two datasets (Nadig HepG2, Nadig Jurkat) and second on two others (Replogle RPE1, Replogle K562), with notably weaker performance on Norman K562. For enrichment factor, scDEBART ranked first on four of five datasets and third on the remaining one. This cross-pipeline validation indicates that scDEBART captures generalizable DEG rankings that extend beyond the scVI-based DE methodology used during training, while cosine similarity is more sensitive to dataset-specific characteristics (Appendix A.6.2).

**Ablation, scaling, and supervision quality analyses.** We organized our analyses around three design axes, each motivated by a distinct hypothesis about what contributes to scDEBART's performance. First, to probe whether large-scale pretraining on expression-change profiles meaningfully contributes to downstream performance, we both removed pretraining entirely and varied the pretraining corpus size; pretraining ablation reduced enrichment factor by 14.8% ($14.9 \rightarrow 12.7$, $p = 6.58 \times 10^{-7}$) and F1 by 9.9% at top-50 DEGs, and downstream performance increased with corpus size (EF 12.51–12.88 at 25–50% vs. 14.33 at 100%), suggesting that pretraining scale is a substantial contributor (Appendix A.7.1, A.8). Second, motivated by the heterogeneity of CRISPR-perturbed cells in Perturb-seq—where incomplete transduction and variable guide efficiency can dilute true effects—we asked whether perturbation-response-based cell filtering improves fine-tuning targets; removing this filter yielded a modest 3.7% EF reduction, indicating a small benefit that does not appear to be the dominant factor (Appendix A.7.1). Third, because the scaling experiments hold the pretraining objective constant and therefore cannot disentangle supervision *quality* from data *quantity*, we asked whether the reliability of scVI-denoised logFC supervision contributes independently: we injected Gaussian noise into scVI-derived logFC targets during fine-tuning, with noise magnitude calibrated to the gene-wise scVI posterior uncertainty so that less certain estimates received proportionally larger perturbations. This produced monotonic but modest decreases across cosine similarity, RMSE, EF, and F1 (Appendix A.9), providing suggestive evidence that uncertainty-aware scVI-denoised supervision

contributes to performance beyond data scale, though a more complete test would require noising the pretraining corpus itself. Together, these analyses suggest that scDEBART's gains arise from the joint contribution of pretraining at sufficient scale, perturbation-level cell quality control, and uncertainty-aware scVI-denoised supervision, rather than from any single factor in isolation.

## 4.2. Reverse Perturbation Identification

We evaluated models on a reverse perturbation task: identifying the perturbed gene(s) from an expression-change profile. Given an observed logFC profile from a held-out perturbation, we generated model-predicted response profiles for a candidate set of perturbations and ranked these candidates by similarity to the observed profile to infer the perturbed gene(s).

Following Cui et al. (2024), we selected the top 20 most frequently perturbed genes in the Norman K562 dataset (after applying filtering in Appendix A.2.2). Among all possible single and double perturbation combinations (210 total: 20 single + 190 double), 57 were observed in the dataset. We split these into 47 train, 3 validation, and 7 test samples, and fine-tuned each model on the training set to predict logFC for all 210 candidates. For scDEBART, candidates are model-predicted logFC profiles; for scGPT and GEARS, we converted predictions to logFC via matched controls (Appendix A.4.4). For each test perturbation, we ranked candidates by Euclidean distance to the observed logFC and report top-$K$ hit rates ($K \in \{1, \ldots, 8\}$) under two criteria: exact match (correct combination in top-$K$) and partial match (at least one ground-truth gene in top-$K$).

Figure 3 shows that scDEBART substantially outperforms baselines under this protocol. scDEBART achieves 71.4% top-1 exact match accuracy (5/7 test samples; 95% bootstrap CI: [0.286, 1.000]), versus 0.0% [0.000, 0.000] for both scGPT and GEARS (Figure 3A). For partial match, scDEBART achieves 100% top-1 accuracy (7/7; 95% CI: [1.000, 1.000]), compared to 42.9% [0.143, 0.714] for GEARS and 0.0% [0.000, 0.000] for scGPT. The wide CI for scDEBART's exact match reflects the small test set size ($n = 7$); nonetheless, the lower bound of the CI (0.286) already exceeds the point estimates of both baselines (0.000). Panel B illustrates this for the MAPK1+ETS2 double perturbation: scDEBART's top-1 prediction correctly identifies the ground-truth pair, while scGPT predicts the single TBX3 perturbation and GEARS predicts CEBPE+ETS2 (partially correct).

## 4.3. Cross-Modal Generalization to Drug Perturbations

To assess whether perturbation-induced regulatory patterns learned from genetic perturbations extend to distinct modalities, we evaluated the Replogle K562-fine-tuned model on

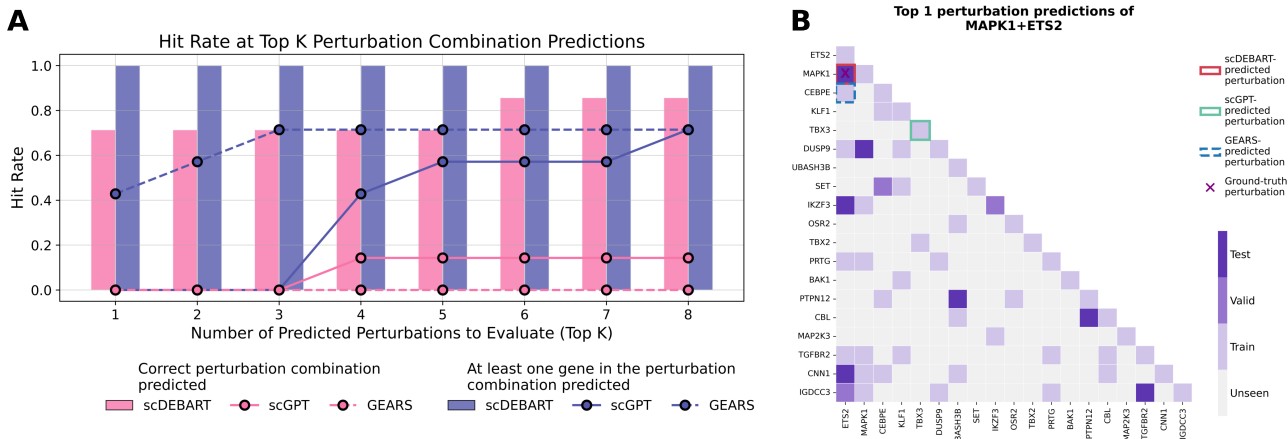

*Figure 3.* **Reverse perturbation identification on Norman K562.** (A) Top-$K$ hit rates for identifying the perturbed gene(s) from 210 candidates (20 single + 190 double) by ranking Euclidean distances to observed logFC profiles. Exact match: correct combination in top-$K$; partial match: at least one correct gene in top-$K$. (B) Heatmap of predicted logFC for the MAPK1+ETS2 test case. scDEBART correctly identifies the ground-truth combination (marked ×) as top-1, whereas scGPT predicts the single TBX3 perturbation and GEARS predicts CEBPE+ETS2 (partially correct).

drug-perturbed cells from SCIPLEX (Srivatsan et al., 2020) without additional training. Drugs inhibit or activate multiple target genes simultaneously via polypharmacological mechanisms (Reddy & Zhang, 2013), and their combined effects manifest as transcriptome-wide expression changes. However, SCIPLEX constitutes a challenging transfer setting: drug-induced responses are typically subtle (IQR FC [0.74, 1.21] for top-20 DEGs) and reflect polypharmacological and off-target effects beyond mapped primary targets. Our goal is not to fully model the complete pharmacological mechanism, but rather to probe whether regulatory patterns learned from genetic perturbations provide signal for drug responses under a primary-target approximation. We focused evaluation on top-20 true DEGs per condition, as these capture the most informative drug-responsive signal. Among 188 compounds, 65 were linked with primary inhibitor targets and subjected to simulated multi-gene inhibition (Methods, Appendix A.5). Since GEARS, MLP and LR have fixed output sizes, we report cross-dataset transfer only for scVI-scGPT among baselines.

Figure 4 summarizes dose-dependent trends. scDEBART showed increasing alignment with drug-induced responses as dose increased: cosine similarity rose from 0.04 (10 nM) to 0.30 (10000 nM), whereas scVI-scGPT showed near-zero to negative cosine similarity across doses (−0.05 to −0.27), indicating limited agreement in predicted response direction. scDEBART achieved moderate DEG recovery (EF 2.91–4.32), while scVI-scGPT remained near random (EF 0.36–0.95). RMSE ranged from 0.41–1.54 for scDE-BART across doses, lower than scVI-scGPT (1.96–2.87). F1 scores were modest for both models (scDEBART 0.05–0.09; scVI-scGPT 0.03–0.24), reflecting the low magnitude of drug-induced logFC changes which fall below the 1.2-fold

classification threshold.

Overall, scDEBART demonstrated partial cross-modal transfer: without drug-specific training, it recovered drug-responsive genes above random expectation and showed improved alignment at higher doses. However, performance remained substantially lower than on genetic perturbations (cosine similarity 0.30 at the highest dose vs. 0.737 on Replogle K562 at top-50), likely reflecting fundamental differences between genetic knockdown and complex drug mechanisms, including dose-dependent nonlinear effects and incomplete target annotations. These findings suggest potential for cross-modal generalization while underscoring the importance of incorporating pharmacological training data for accurate drug-response prediction.

## 5. Conclusion

We introduced scDEBART, a BART transformer-based model for predicting single-cell perturbation responses, built around two explicit design strategies. First, scDE-BART is pretrained to directly predict log fold-changes conditioned on basal expression, encouraging the model to learn context-dependent regulatory responses and thereby infer perturbation-induced transcriptional changes. Second, we derive expression changes from scVI-denoised expression (rather than from point predictions that are sensitive to dropout and zero inflation) and restrict training to reliably detected genes, so that the learned log fold-changes are less dominated by technical zeros.

Across five large-scale Perturb-seq datasets from three independent studies, scDEBART demonstrated strong performance across multiple metrics, with the most pronounced

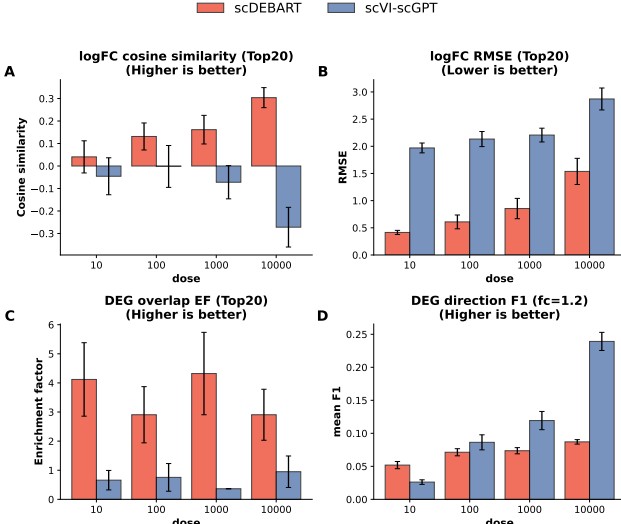

*Figure 4.* **Cross-modal transfer to drug perturbations in SCI-PLEX.** A model fine-tuned on genetic perturbations (Replogle K562) was evaluated on SCIPLEX drug responses across four doses (10, 100, 1000, 10000 nM). Drugs were simulated as multi-gene knockdowns by mapping compounds to primary protein targets (Methods), retaining 65 of 188 compounds with at least one target gene in the model vocabulary (Appendix A.5). Metrics are computed on top-20 DEGs per drug condition: (A) cosine similarity, (B) RMSE, (C) enrichment factor, and (D) F1 score. Error bars indicate 95% confidence intervals across drugs at each dose.

gains in DEG recovery (enrichment factor) and direction classification (F1 score), and competitive cosine similarity that ranked first or second on all datasets. In addition, dropout-focused analyses suggest that several baseline models tend to prioritize genes with low detection rates among their top predicted DEGs, which can exhibit distorted fold-changes due to technical zeros.

We also observed preliminary evidence of cross-modal transfer to pharmacological perturbations in SCIPLEX: without drug-specific training, scDEBART achieved dose-dependent improvements and recovered drug-responsive genes above random expectation, although performance remained substantially below that on genetic perturbations. This gap likely reflects fundamental differences between genetic knockdown and indirect drug mechanisms, incomplete target annotations, and dose-dependent nonlinearities not captured by a knockdown-style input.

Several limitations suggest concrete directions for follow-up work. First, our supervision derives from scVI-based differential expression, which may introduce inductive biases from this generative model. Future work could explore alternative denoising methods or ensemble approaches to mitigate potential biases while maintaining computational efficiency. Second, restricting training to reliably detected genes improves robustness but reduces coverage of lowly expressed or rare genes. This trade-off could be

addressed through gene-wise uncertainty modeling or hierarchical priors. Third, cross-modal transfer on drugs remains limited by incomplete target annotations and variable off-target/polypharmacological effects across compounds, suggesting that improved target and perturbation representations will be critical for accurate pharmacological response prediction. Finally, while this study demonstrates per-dataset performance for fine-tuning, developing a unified perturbation prediction model across diverse cell types and perturbation gene spaces would enable broader applications, including therapeutic target discovery and gene function annotation. Overall, our results demonstrate that dropout-robust supervision and large-scale learning of expression dynamics provide a promising foundation for perturbation-effect modeling.

## Software and Data

All code and processed datasets required to reproduce the experiments are publicly available. The code for training and inference of scDEBART is available at `https://github.com/Jieun-Sung/scDEBART`, and the processed datasets used in this study are hosted on HuggingFace at `https://huggingface.co/datasets/Jieun-S/scDEBART`.

Detailed information on all datasets used in this study, including accession numbers and download sources, is provided in Appendix A.11.

## Acknowledgements

This work was supported by the National Research Foundation of Korea (NRF) under Grant RS-2024-00411145. Computational resources and technical support were provided by the Korea Bio Data Station (K-BDS).

## Impact Statement

This work advances machine learning for single-cell perturbation prediction, with potential applications in drug discovery and therapeutic target identification. By enabling accurate *in silico* prediction of cellular responses to genetic perturbations, scDEBART may accelerate early-stage drug development and facilitate functional genomics studies, potentially reducing the cost and time required for experimental validation. However, we acknowledge several considerations: (i) pretraining requires substantial computational resources (6.28M DE profiles from 66.6M cells), which may limit accessibility for resource-constrained laboratories, potentially exacerbating existing disparities in biomedical research infrastructure; (ii) the model's predictions, while validated across multiple datasets, remain approximate and should not be used as a substitute for experimental valida-

tion; and (iii) genetic perturbation prediction tools could be misapplied in synthetic biology or bioengineering contexts in ways not anticipated by the authors. To mitigate these concerns, we prioritize reproducibility through detailed methodology, plan to release code and pretrained weights, and encourage users to validate predictions experimentally before making critical decisions.

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

# A. Appendix

## A.1. Construction of Differential Expression Corpus from CELLxGENE

### A.1.1. DATA COLLECTION AND QUALITY CONTROL

We curated 1,573 datasets (893 human) from the CELLxGENE data portal (accessed January 30, 2025), comprising over 106 million cells. We applied two dataset-level inclusion criteria: (i) the presence of normal (healthy) human cell populations to avoid learning purely disease-specific artifacts, and (ii) a dataset size between 10,000 and 1,000,000 cells to balance computational efficiency with statistical stability. This filtering yielded 617 datasets comprising 66,611,859 cells spanning 61 tissue types, 81 disease types, and 23 sequencing technologies.

For each dataset, we retained protein-coding genes only and applied adaptive cell-level quality control based on total UMI counts and mitochondrial read fraction. Specifically, we kept cells whose total UMI counts and mitochondrial read percentages fell within $\pm 3$ standard deviations of the dataset-specific mean, following the protocol in GeneFormer (Theodoris et al., 2023).

The resulting dataset corpus is enriched for brain (31.4%) and blood (10.6%) tissues, and normal cells comprise 81.9% of all cells. Most cells (79.5%) were generated using 10x Genomics 3' v3 and v2 platforms (Appendix Figure 5).

### A.1.2. CLUSTERING WITH SECUER

After quality control, we identified transcriptionally coherent cell populations within each dataset. Raw counts were normalized to 10,000 total counts per cell, log-transformed as $\log(1 + x)$, and the top 2,000 highly variable genes (HVGs) were selected using Scanpy's dispersion-based method. We then scaled HVG expression values (clipped at a maximum of 10) and computed PCA embeddings for clustering.

We applied SECUER (Wei et al., 2022), a density-based clustering algorithm for single-cell data, to the PCA representation. SECUER was configured with $K_{\mathrm{nn}} = 2$ for k-nearest neighbor graph construction. Unlike resolution-based methods (e.g., Leiden), SECUER adaptively determines the optimal number of clusters based on local density structure, enabling the identification of biologically meaningful populations without manual resolution tuning. Across the 617 datasets, this procedure identified 98,753 clusters with a median size of 278 cells (Appendix Figure 6A). The number of clusters increases with dataset size and shows diminishing returns for large datasets (Appendix Figure 6B).

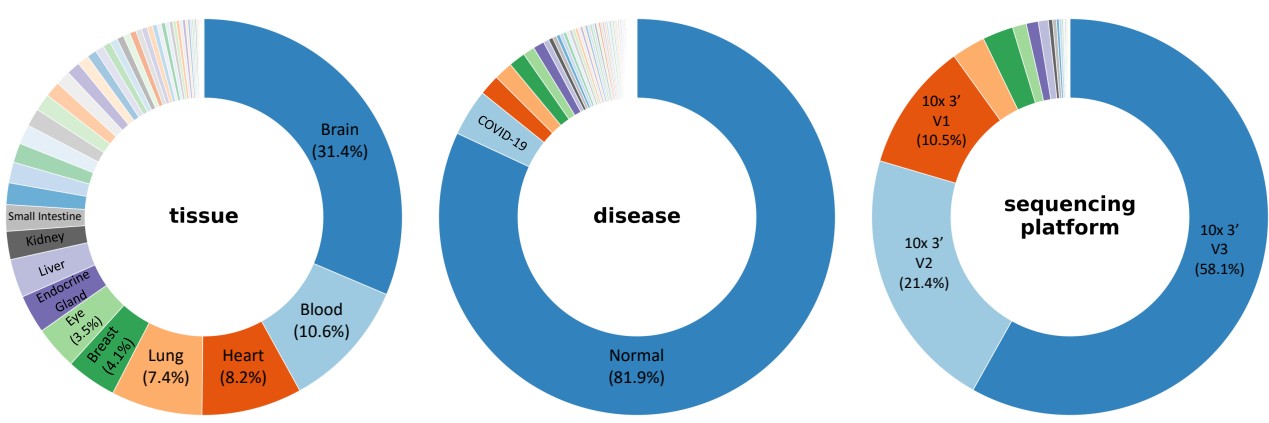

*Figure 5.* **Distribution of cell metadata across the quality-filtered scRNA-seq corpus.** We retained 617 human datasets comprising approximately 66.6 million cells. The charts summarize the composition by: (Left) tissue of origin, with largest contributions from brain (31.4%), blood (10.6%), and heart (8.2%); (Center) disease status, showing a predominance of normal cells (81.9%); and (Right) sequencing platform, with most data generated using 10x Genomics 3' chemistry.

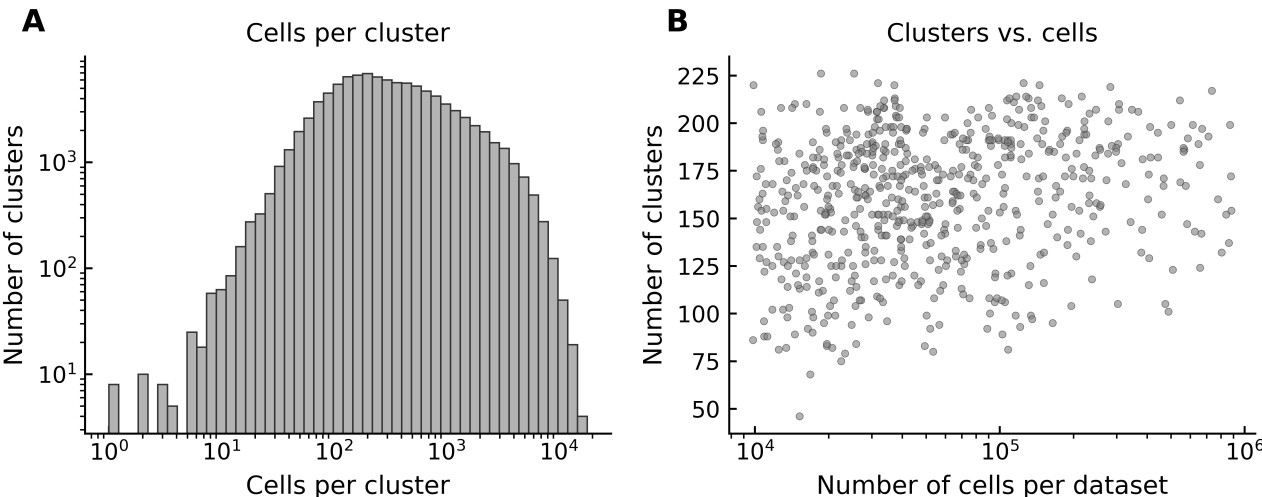

*Figure 6.* **Overview of cell populations identified by SECUER clustering. (A)** Distribution of cluster sizes, with a median of 278 cells per cluster. **(B)** Number of clusters versus dataset size.

### A.1.3. SCVI MODEL TRAINING

For each dataset, we trained an scVI model (Lopez et al., 2018) on the filtered cells to learn a latent representation that captures biological variation while accounting for technical confounders such as batch effect and dropout. The model was configured with 64 latent dimensions and used cell-type annotations as auxiliary labels when available. We trained scVI with a batch size of 512 for up to 200 epochs, using early stopping with a patience of 5 epochs. All models were trained with a fixed random seed (42) to ensure reproducibility.

### A.1.4. CLUSTER PAIR SELECTION

After training, we extracted 64-dimensional latent representations for all cells and computed cluster centroids as the mean latent vector of cells within each cluster. To select informative cluster pairs for differential expression analysis, we computed pairwise cosine similarities between all cluster centroids and retained pairs whose similarities fell within the 10th–90th percentile range. This filtering removes (i) nearly identical clusters that would yield trivial DE profiles and (ii) highly dissimilar clusters that may reflect multi-step trajectories rather than direct regulatory transitions.

To reduce computational cost and avoid over-representing datasets with an exceptionally large number of cluster pairs, if the number of retained pairs exceeded 100,000, we uniformly subsampled 100,000 pairs using a fixed random seed (42).

### A.1.5. DIFFERENTIAL EXPRESSION CALCULATION

For each selected cluster pair $(C_1, C_2)$, we performed differential expression analysis using scVI's posterior sampling framework by drawing 4,000 samples from the posterior distribution of gene expression and running the test in `change` mode with an effect-size threshold of $\delta = 0.25$. The reference cluster was randomly assigned, as our goal was to capture gene sets and fold-change magnitudes rather than directional comparisons. We used a batch size of 256 and disabled batch correction, as the scVI latent representation already accounts for technical confounders. Outlier-cell filtering was not applied to preserve biological heterogeneity.

For each gene $g$, we computed:

**(i) Probability of differential expression (Proba-DE):** The posterior probability that gene $g$ is differentially expressed between $C_1$ and $C_2$, estimated as

$$P(\text{DE}_g \mid \mathbf{X}) = \mathbb{E}_{z \sim q}\left[\mathbb{1}\left(|\log_2(\rho_{g,C_2}/\rho_{g,C_1})| > 0.25\right)\right]$$

where $\rho_{g,C}$ is the expected normalized expression of gene $g$ in cluster $C$ under the ZINB generative model, and $q$ is the variational posterior over latent variables.

**(ii) Posterior-averaged normalized expression (scVI-denoised expression):** The average normalized expression in each cluster,

$$\mu_{g,C} = \frac{1}{S} \sum_{s=1}^{S} \rho_{g,C}^{(s)}$$

where $S = 4,000$ is the number of posterior samples.

**(iii) Posterior median log fold-change (logFC):** The median of sampled log fold-changes,

$$\text{logFC}_g = \text{median} \left\{ \log_2 \left( \frac{\rho_{g,C_2}^{(s)} + \epsilon}{\rho_{g,C_1}^{(s)} + \epsilon} \right) \right\}_{s=1}^{4000}$$

where $\rho_{g,C}^{(s)}$ denotes the $s$-th posterior sample of normalized expression (library size 10,000) and $\epsilon = 10^{-8}$ prevents division by zero.

**(iv) Non-zero proportion:** The fraction of cells in cluster $C$ with detected (non-zero) UMI counts for gene $g$,

$$p_{g,C} = \frac{1}{|C|} \sum_{i \in C} \mathbb{1}(x_{ig} > 0)$$

where $x_{ig}$ is the raw count for gene $g$ in cell $i$. This serves as a proxy for gene detection rate and technical dropout severity.

### A.1.6. RATIONALE FOR FILTERING THRESHOLDS

A fundamental challenge in large-scale DE analysis is the absence of universal biological criteria for defining differentially expressed genes—appropriate thresholds vary across datasets due to differences in cell types, experimental protocols, and biological contexts. We leveraged three statistical measures computed by scVI's differential expression framework: non-zero proportion, FC, and Proba-DE. Using 100 randomly sampled datasets from our collection of 617, we performed a grid search across Proba-DE (0.5–0.9), non-zero proportion (0.1–0.3), and fold-change (1.5–3.0) thresholds, computing retained DE gene counts for each configuration (Appendix Figure 7). The red dashed line marks our target of 1,024 genes per profile, chosen to capture highly responsive genes while maintaining computational tractability for transformer models.

We first applied the most stringent non-zero proportion threshold to minimize dropout-related false positives, then optimized Proba-DE and fold-change thresholds empirically. Specifically, we selected non-zero proportion $\geq 0.3$ as the most conservative threshold, requiring genes to be detected in at least 30% of cells in both clusters. Within this constraint, we chose Proba-DE $\geq 0.8$ and fold-change $\geq 1.5$, as this combination consistently yielded median DEG counts near our 1,024-gene target. These thresholds ensure strong posterior evidence of differential expression while excluding marginal changes within the range of technical and biological noise.

We applied these filtering criteria to retain high-confidence DE genes:

- Non-zero proportion $\geq 0.3$ in both clusters: Ensures genes are reliably detected and not dominated by dropout

- Proba-DE $\geq 0.8$: High posterior confidence of differential expression

- Absolute fold-change $\geq 1.5$ ($\log_2$ scale $\geq 0.585$): Excludes negligible expression changes

### A.1.7. GLOBAL CHARACTERISTICS OF DE PROFILES

After applying DE filtering criteria, profiles with fewer than 10 genes were discarded as insufficiently informative. For remaining profiles, we selected up to $K = 1024$ genes ranked by absolute logFC to focus on the most responsive genes while maintaining computational tractability. This curation procedure yielded 6,282,796 high-quality DE profiles from 617 datasets, covering 17,944 genes with a median of 10,611 profiles per dataset.

To characterize the statistical properties of curated DE profiles, we randomly sampled 10% of profiles (628,280 profiles) and examined the distributions of log fold-changes, normalized gene expression levels, and the number of genes per profile (Appendix Figure 8). Log fold-changes exhibit a distribution centered near zero (median $= -0.62$, IQR $= [-1.19, 1.15]$),

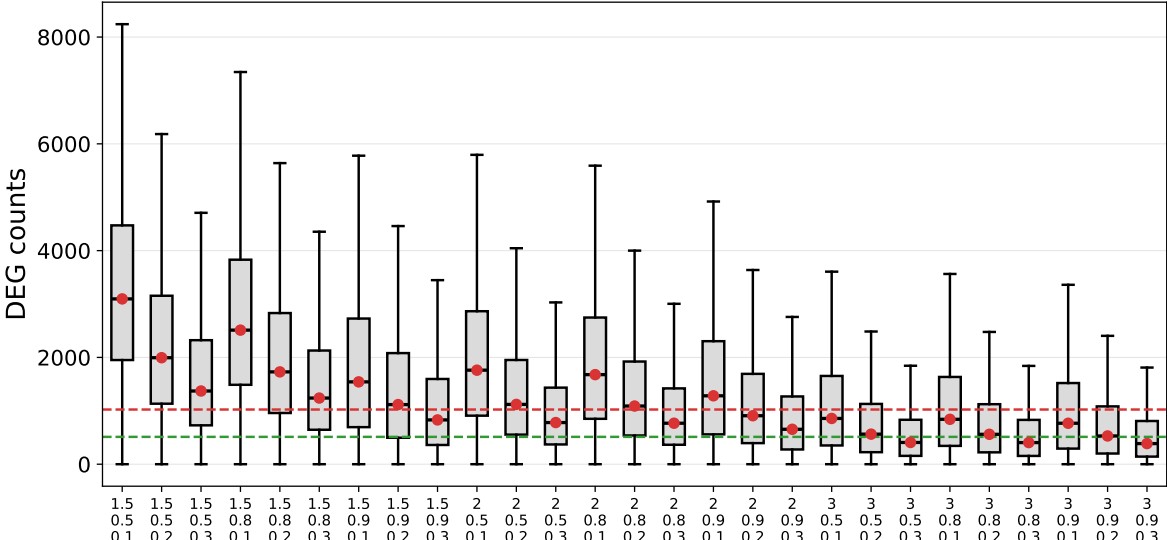

*Figure 7.* **Impact of filtering thresholds on DE gene counts.** Boxplots show the distribution of DEG counts across cluster pairs for 100 randomly sampled datasets under different combinations of fold-change (FC), Proba-DE (PD), and non-zero proportion (NZ) thresholds. Each x-axis tick represents a parameter combination (from top to bottom: FC, PD, NZ). Red dots indicate median values for each configuration. The red dashed line marks our chosen target of 1,024 genes per profile (used in this study), and the green dashed line marks an alternative 512-gene target shown for reference. The selected thresholds (FC $\geq$ 1.5, NZ $\geq$ 0.3, PD $\geq$ 0.8) balance stringent dropout mitigation with optimal gene coverage closest to the 1,024-gene target.

with most changes falling within a biologically interpretable range while extreme fold-changes extend to $\pm 13$ (Appendix Figure 8A). scVI-denoised expression levels (sum = 1e+4) in baseline populations display a characteristic right-skewed distribution (median = 1.43, IQR = $[0.69, 3.22]$), reflecting the prevalence of lowly expressed genes in scRNA-seq data, with a long tail extending to highly expressed genes (Appendix Figure 8B). The number of genes per DE profile shows substantial variability (median = 505, IQR = $[238, 964]$), with most profiles containing hundreds of DE genes and a subset reaching the imposed maximum of 1,024 genes (Appendix Figure 8C).

To assess whether DE profiles capture tissue-specific biological structure, we performed PCA on the full collection of 6.28M profiles using log fold-changes as features (Appendix Figure 9). Notably, brain-derived profiles form a tight, cohesive cluster, reflecting the highly conserved transcriptional programs and cellular compositions across brain regions and datasets. In contrast, other tissues such as lung, blood, and small intestine display more dispersed distributions in transcriptomic space, suggesting greater heterogeneity in cellular compositions, developmental stages, or functional states sampled across studies.

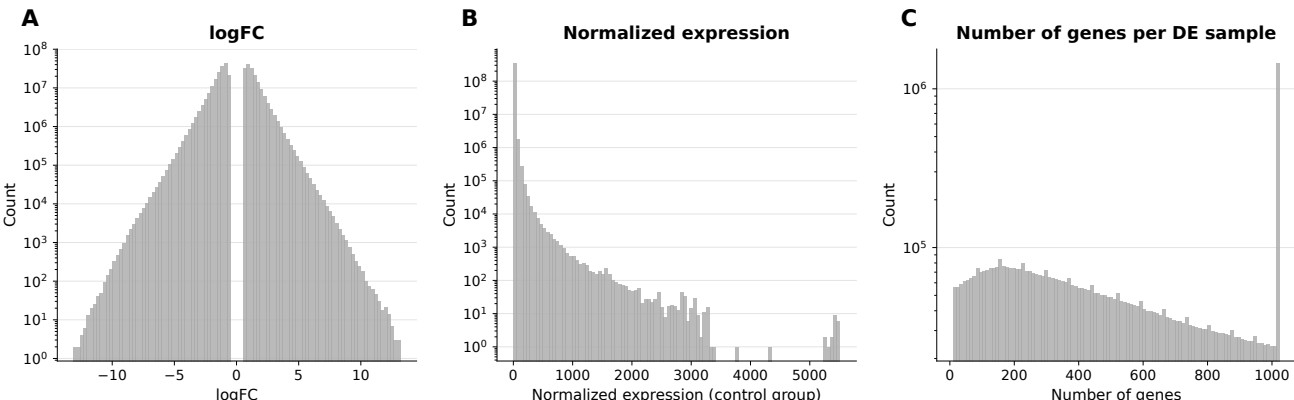

*Figure 8.* **Global characteristics of curated DE profiles.** Distributions shown for a 10% random sample (628,280 profiles) of the full collection. **(A)** Log fold-change distribution is centered near zero, with most changes within $\pm 5$. **(B)** scVI-denoised gene expression in baseline populations follows a right-skewed distribution typical of scRNA-seq data. **(C)** Number of genes per profile.

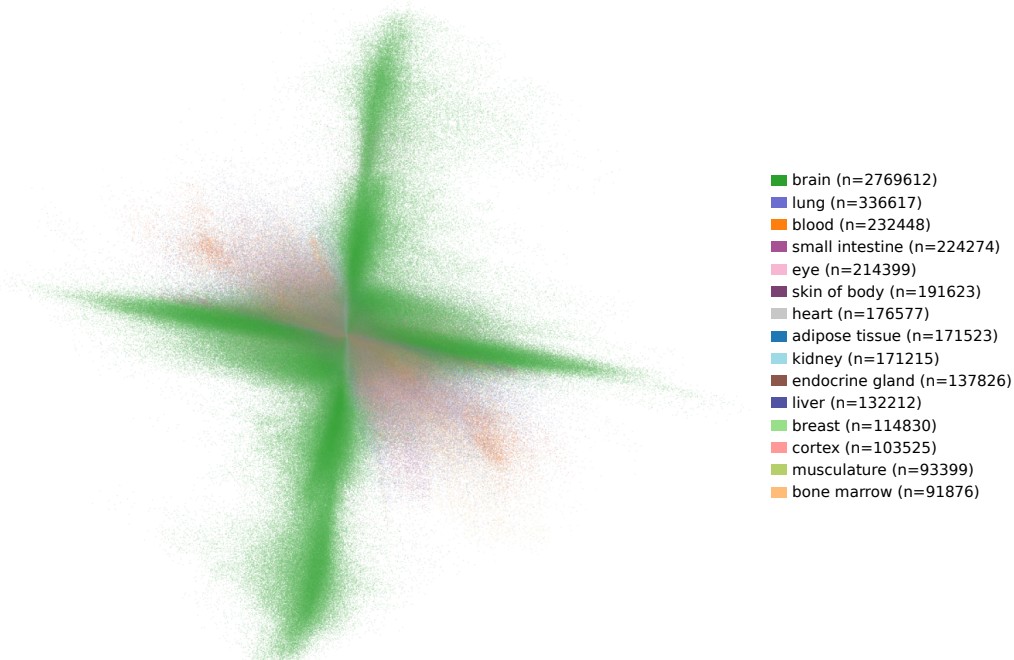

*Figure 9.* **Tissue-specific organization of DE profiles in transcriptomic space.** PCA projection of all 6.28M DE profiles colored by tissue of origin. Each point represents a single DE profile, with the top 15 tissues by profile count shown. Profiles cluster predominantly by tissue type, demonstrating that our curation pipeline captures biologically meaningful transcriptional signatures rather than technical artifacts. Notably, brain-derived profiles (2.77M) form a tight, cohesive cluster, reflecting highly conserved transcriptional programs across brain regions and datasets, while other tissues such as lung (336K), blood (232K), and small intestine (224K) display more dispersed distributions, suggesting greater heterogeneity in cellular compositions and functional states across studies.

## A.2. Computation of Gene Perturbation Effects from Perturb-seq

### A.2.1. PERTURBATION STRENGTH QUANTIFICATION AND FILTERING IN PERTURB-SEQ

Perturb-seq technologies (Dixit et al., 2016) enable massively parallel functional genomics by coupling CRISPR-based perturbations with single-cell RNA sequencing. In practice, however, cells assigned to the same perturbation can exhibit heterogeneous responses due to incomplete viral transduction, variable guide RNA expression, and differences in CRISPR interference/editing efficiency. This heterogeneity can dilute true biological effects and complicate downstream modeling if weakly perturbed or effectively unperturbed cells are not properly identified and filtered.

To address this challenge, Song et al. (Song et al., 2025) introduced the perturbation-response score (PS), which quantifies perturbation strength at the single-cell level based on downstream transcriptional response. Unlike binary approaches such as mixscape (Papalexi et al., 2021), PS assigns a continuous score between 0 (little evidence of perturbation) and 1 (strong inferred perturbation effect), better reflecting partial perturbations commonly observed in CRISPRi, dose-titrated knockdowns, and variable knockout efficiencies. Importantly, the PS framework highlights that overlooking response heterogeneity—and thus retaining weakly perturbed cells in treated groups—can bias estimated effects and obscure mechanistic signals, yet such filtering is not consistently emphasized across prior perturbation-modeling studies.

Motivated by this perspective, our work incorporates stringent quality control when constructing training targets: we leverage PS-informed filtering (where available) and related response-based criteria to curate cell groups whose perturbation signals are sufficiently strong and coherent, thereby reducing contamination from effectively unperturbed cells when learning perturbation-induced transcriptional changes.

### A.2.2. FILTERING PERTURB-SEQ DATASETS

We applied a four-step sequential filtering pipeline to Perturb-seq datasets to ensure high-quality perturbation effects (Appendix Table 1).

**Step 1: Quality Control Filtering.** All Perturb-seq datasets underwent the same initial quality control procedures as the CELLxGENE corpus, including adaptive filtering based on total UMI counts and mitochondrial read fraction to remove low-quality cells.

**Step 2: Perturbation Target-Specific Expression Filtering.** We filtered cells based on target gene expression to ensure successful perturbation. For CRISPRi experiments (gene silencing), we retained only cells where the targeted gene showed zero expression, indicating successful knockdown. For CRISPRa experiments (gene activation), we retained only cells where the targeted gene exhibited non-zero expression, confirming successful transcriptional activation.

**Step 3: Perturbation Score (PS) Threshold Filtering.** We utilized the Perturbation Score (PS) (Song et al., 2025) to quantify the confidence of perturbations in each cell. To determine dataset-specific PS thresholds, we swept PS values from 0.00 to 0.75 and selected the cutoff that maximized the separability between perturbed and unperturbed populations in PCA space, as quantified by the Calinski-Harabasz score (Appendix Figure 10, top row). In cases where the Calinski-Harabasz score peaked at highly restrictive thresholds ($> 0.5$), we performed a cell-cell similarity $t$-test in normalized gene expression space. For each perturbed gene with at least 5 cells, we computed pairwise cosine similarities between cells sharing that perturbation (sampling up to 10,000 pairs per gene). These within-perturbation similarities were compared to a baseline distribution of cosine similarities between randomly sampled cell pairs across the entire filtered dataset (10,000 pairs) using a two-sample Welch's $t$-test. Lower $p$-values indicate that perturbed cells exhibit stronger transcriptional coherence than expected by chance (Appendix Figure 10, middle row). The optimized PS cutoffs for each dataset are shown in Appendix Table 1.

**Step 4: Cluster Coherence Filtering.** After PS-based filtering, we performed cluster coherence analysis to retain only perturbed cells forming compact, well-separated clusters distinct from unperturbed populations. For each perturbed gene, we computed the centroid of filtered perturbed cells in PCA space and measured the Euclidean distance from each cell to both its gene-specific centroid and the unperturbed population centroid. Cells closer to their own gene centroid than to the unperturbed centroid were retained. To exclude outlier-dominated or overly dispersed perturbation groups, we computed a gene-specific spread metric as the 90th percentile of within-cluster distances and filtered perturbation groups whose spread exceeded $\mu + 2\sigma$, where $\mu$ and $\sigma$ are the mean and standard deviation across all genes. Only perturbation groups with at least 5 cells surviving this filter were retained for downstream analysis.

The cumulative effect of this four-step pipeline is shown in Appendix Table 1, with final retention rates ranging from 31.31% (Replogle RPE1) to 54.93% (Nadig HepG2). Visual inspection of PCA embeddings illustrates our filtering strategy (Appendix Figure 11). Initially, perturbed cells are widely dispersed throughout the control manifold. Notably, cells with low PS values (light orange) are located near or within the unperturbed cluster, suggesting weak or incomplete perturbation effects that closely resemble control cell states. Our two-stage filtering approach addresses this: Step 3 (PS thresholding) first removes these low-confidence cells, while Step 4 (cluster coherence filtering) subsequently eliminates outliers even among high-PS cells to promote tighter, more homogeneous perturbation-specific clusters. The combined application of these filters progressively compacts perturbed cells into more well-defined groups.

| | Perturb type | PS cutoff | Raw data cells | 1) Filtered low-quality cells | 2) Perturb gene expr | 3) PS score $\geq$ cutoff | 4) Close clusters only | Final perturb genes |
|---|---|---|---|---|---|---|---|---|
| Norman K562 | CRISPRa | 0.35 | 91,205 | 89,940 | 62,065 | 49,096 | **40,500** | 221 |
| Replogle K562 | CRISPRi | 0.25 | 310,385 | 306,865 | 281,901 | 227,582 | **157,676** | 2,065 |
| Replogle RPE1 | CRISPRi | 0.45 | 247,914 | 243,811 | 175,999 | 101,558 | **77,614** | 1,564 |
| Nadig Jurkat | CRISPRi | 0.2 | 262,956 | 250,943 | 189,240 | 151,225 | **102,251** | 1,738 |
| Nadig HepG2 | CRISPRi | 0.3 | 145,473 | 142,521 | 129,935 | 110,857 | **79,915** | 2,559 |

*Table 1.* **Sequential filtering steps applied to Perturb-seq datasets.** Numbers indicate cell counts remaining after each quality control step. Step 1: Quality control filtering. Step 2: Perturbation Target-Specific Expression Filtering. Step 3: PS score filtering with dataset-optimized cutoffs. Step 4: Cluster coherence filtering. Percentages in column 4 represent the fraction of raw cells retained after all filtering criteria. The "Final perturb genes" column shows the number of unique perturbed genes retained after filtering.

A.2.3. DIFFERENTIAL EXPRESSION WITH PERTURBATION-AWARE SCVI

Following final cell selection, we applied the same preprocessing pipeline as for CELLxGENE datasets: total-count normalization, log-transformation, and scaling. We trained a separate scVI model for each dataset, configured identically to the CELLxGENE pretraining setting but using the perturbed gene identity as the batch key to explicitly model perturbation-specific variation within the latent space. To manage computational costs, we selected the top 5,000 highly variable genes

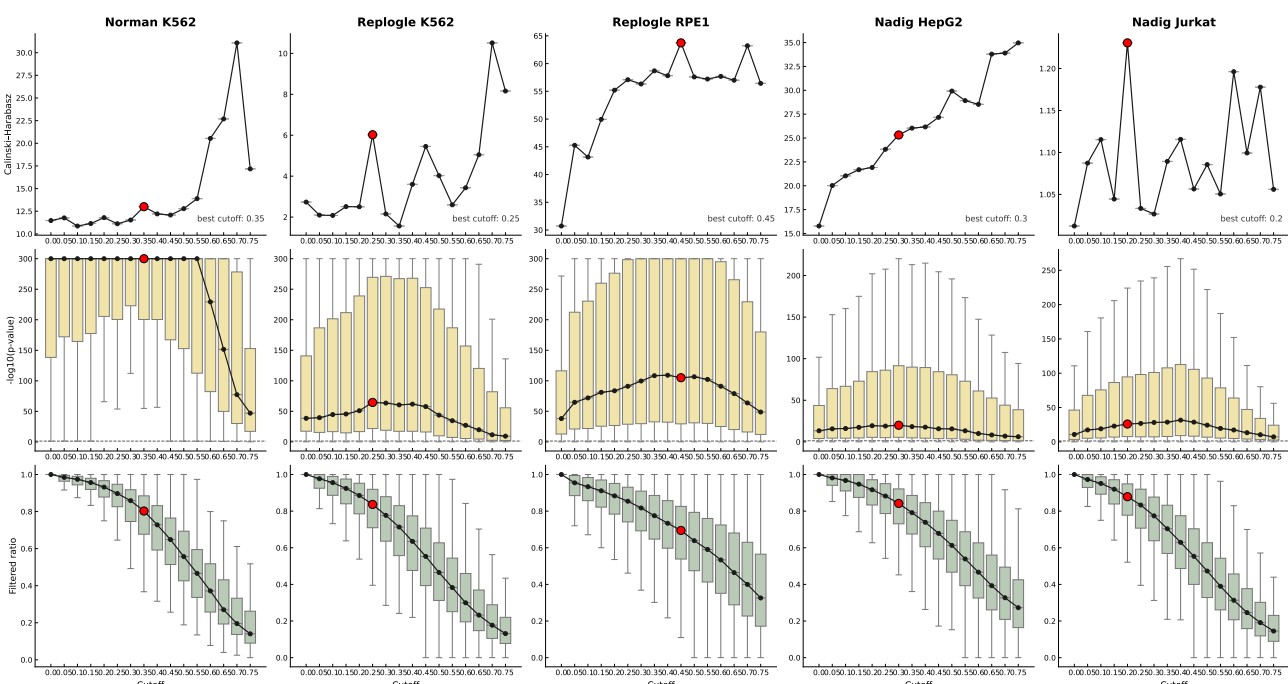

*Figure 10.* **Optimization of Perturbation Score (PS) thresholds for Step 3 filtering.** Each column corresponds to a dataset. **(Top)** Calinski-Harabasz score measuring separation. **(Middle)** Statistical significance from permutation-based $t$-test. **(Bottom)** Cell retention fraction. Red dots indicate selected optimal cutoff.

(HVGs) from scVI-denoised expression for datasets with more than 5,000 genes; otherwise, all genes were retained. Unlike the pretraining corpus, which applies stringent Proba-DE and fold-change thresholds to focus on high-confidence differential expression, fine-tuning retains all 5,000 HVGs to enable the model to predict both strongly and weakly responsive genes. This design allows scDEBART to distinguish true non-responsive genes from technical zeros, a critical capability for perturbation prediction tasks where many genes exhibit minimal or no transcriptional response.

For each perturbed gene $g_{\text{pert}}$, we performed differential expression analysis comparing cells with perturbation of $g_{\text{pert}}$ against unperturbed control cells using the trained scVI posterior sampling framework. To balance computational efficiency with data augmentation for train/validation/test splits, we modified the sampling strategy from the CELLxGENE pipeline: instead of drawing 4,000 posterior samples once per cluster pair, we drew 1,000 posterior samples and repeated this process 50 times independently for each perturbation. This yields 50 distinct profile instances per perturbed gene, where each instance contains posterior-averaged normalized expression (scVI-denoised expression) (control and perturbed) and posterior median log fold-change (logFC) computed from 1,000 samples. We also computed Proba-DE and non-zero proportions for all genes.

Unlike the CELLxGENE pipeline where we applied three filtering criteria (non-zero proportion, fold-change, and Proba-DE), we retained only the non-zero proportion threshold for Perturb-seq profiles to preserve genes with minimal or no expression changes. The model needs to predict both differentially expressed and non-responsive genes to fully capture perturbation effects, so we discarded fold-change and Proba-DE filters that would exclude unchanged genes. We maintained the non-zero proportion threshold because it addresses technical bias from dropout rather than biological signal. Specifically, if the non-zero proportion in either cluster fell below 0.3, the logFC for that gene was masked (set to 0) to minimize dropout-induced technical artifacts, while the gene was retained in the profile to preserve the complete gene set.

### A.3. Pretraining Architecture Details

#### A.3.1. ENCODING GENE PRIOR KNOWLEDGE

**BioBERT-based functional embeddings.** We generated contextualized gene embeddings from textual functional descriptions using BioBERT ([Lee et al., 2020](#)). Gene summaries were retrieved via MyGene.info API ([Lelong et al., 2022](#)). We used

BioBERT-Base v1.1 (12 layers, hidden size 768) and extracted the final-layer [CLS] representation as a 768-dimensional embedding.

**Pathway membership encoding.** We encoded pathway membership using curated gene sets from MSigDB v2024.1 C2 (Liberzon et al., 2011), covering 7,411 pathways and 21,290 genes. For each gene, we constructed a binary membership vector $\mathbf{r}_i \in \{0, 1\}^{7411}$.

### A.3.2. COMPLETE MODEL ARCHITECTURE

The complete scDEBART architecture consists of the following components.

**Input Embedding Layer.** For each gene $g_i$ with logFC value $x_i$ and baseline scVI-denoised expression $\mu_i$, we construct the encoder input embedding as follows:

- **Pathway embedding projection:** The binary pathway membership vector $\mathbf{r}_i \in \{0, 1\}^{7411}$ is projected via $W_r \in \mathbb{R}^{768 \times 7411}$.

- **BioBERT embedding:** $\mathbf{b}_i \in \mathbb{R}^{768}$ (precomputed; no additional projection).

- **Knowledge fusion:** The pathway and BioBERT embeddings are concatenated and projected: $W_k[W_r\mathbf{r}_i; \mathbf{b}_i]$, where $W_k \in \mathbb{R}^{768 \times 1536}$.

- **LogFC embedding:** $W_{\text{logFC}}x_i$, where $W_{\text{logFC}} \in \mathbb{R}^{768 \times 1}$.

- **Expression embedding:** $W_{\text{expr}} \log(1 + \mu_i)$, where $W_{\text{expr}} \in \mathbb{R}^{768 \times 1}$.

- **Final gene representation:**

$$\mathbf{h}_i = W_p[W_k(W_r\mathbf{r}_i; \mathbf{b}_i); W_{\text{logFC}}x_i; W_{\text{expr}} \log(1 + \mu_i)],$$

  where $W_p \in \mathbb{R}^{768 \times 2304}$ produces a 768-dimensional input embedding for the encoder.

**Encoder Stack (8 layers).** The encoder consists of $N_{\text{enc}} = 8$ Transformer layers with model dimension $d = 768$:

- **Multi-head self-attention:** 12 heads, head dimension 64.

- **Attention implementation:** Flash Attention via scaled_dot_product_attention when available.

- **FFN:** $\text{FFN}(\mathbf{h}) = W_2 \cdot \text{GELU}(W_1\mathbf{h})$ with hidden dimension $4d = 3072$, i.e., $W_1 \in \mathbb{R}^{3072 \times 768}$ and $W_2 \in \mathbb{R}^{768 \times 3072}$.

- **Residual/LN/Dropout:** pre-norm with residual connections and dropout rate 0.2.

The final encoder outputs are denoted $\mathbf{z}_i$.

**Decoder Stack (4 layers).** The decoder consists of $N_{\text{dec}} = 4$ Transformer layers:

- **Masked self-attention:** causal masking, 12 heads, head dimension 64.

- **Cross-attention:** attends to encoder outputs $\mathbf{z}_i$, 12 heads, head dimension 64.

- **FFN:** same as the encoder (768→3072→768, GELU).

- **Residual/LN/Dropout:** dropout rate 0.2.

The final decoder hidden state for gene $g_i$ is denoted $\mathbf{d}_i$.

**Prediction Heads.** Two lightweight MLP heads are applied:

- **Embedding reconstruction head:**
$$\hat{\mathbf{h}}_i = W_{\text{rec2}} \cdot \text{GELU}(W_{\text{rec1}} \mathbf{d}_i),$$
where $W_{\text{rec1}} \in \mathbb{R}^{1536 \times 768}$ and $W_{\text{rec2}} \in \mathbb{R}^{768 \times 1536}$.

- **LogFC prediction head:**
$$\hat{x}_i = W_{\text{out2}} \cdot \text{GELU}(W_{\text{out1}} \hat{\mathbf{h}}_i),$$
where $W_{\text{out1}} \in \mathbb{R}^{512 \times 768}$ and $W_{\text{out2}} \in \mathbb{R}^{1 \times 512}$.

**Pretraining Objective and Masking.** For each DE profile containing $N$ genes, we select a masked set $\mathcal{M} \subset \{1, \ldots, N\}$ by sampling approximately 15% of non-padded genes. For $i \in \mathcal{M}$, we set the input logFC scalar to zero and overwrite the corresponding logFC embedding with an all-zero vector, while keeping gene priors and baseline expression unchanged. The model reconstructs targets only at masked positions:

$$\mathcal{L}_{\text{total}} = \alpha \cdot \mathcal{L}_{\text{logFC}} + (1 - \alpha) \cdot \mathcal{L}_{\text{emb}},$$

where

$$\mathcal{L}_{\text{logFC}} = \frac{1}{|\mathcal{M}|} \sum_{i \in \mathcal{M}} (x_i - \hat{x}_i)^2, \quad \mathcal{L}_{\text{emb}} = \frac{1}{|\mathcal{M}|} \sum_{i \in \mathcal{M}} \left\| \mathbf{h}_i - \hat{\mathbf{h}}_i \right\|_2^2,$$

with $\alpha = 0.8$. Here, $\mathbf{h}_i$ denotes the clean (unmasked) input embedding for gene $g_i$.

### A.3.3. TRAINING HYPERPARAMETERS

scDEBART was pretrained with the configuration in Appendix Table 2.

*Table 2.* Hyperparameter configuration for scDEBART pretraining.

| Hyperparameter | Value |
|---|---|
| Optimizer | AdamW ($\beta_1 = 0.9$, $\beta_2 = 0.999$) |
| Learning rate | $5 \times 10^{-5}$ |
| Weight decay | $10^{-3}$ |
| Gradient clipping | 0.5 (max norm) |
| Batch size (total) | 340 (85 per GPU, 4 GPUs) |
| Max epochs | 50 |
| Early stopping patience | 5 |
| Training stopped at epoch | 22 |
| Dropout rate | 0.2 |
| Mask probability | 0.15 |
| Loss weighting ($\alpha$) | 0.8 (logFC) + 0.2 (embedding) |
| Warmup steps | 5,000 |
| Mixed precision | bfloat16 |
| Hardware | $4 \times$ NVIDIA H100 (80GB) |

Training used DeepSpeed ZeRO Stage 3 with CPU offloading and activation checkpointing, taking approximately 113 hours of wall-clock time on 4 GPUs (452 GPU-hours).

### A.4. Fine-tuning on Perturb-seq Datasets

#### A.4.1. MODEL ARCHITECTURE AND FORWARD PASS

For fine-tuning on genetic perturbation prediction tasks, scDEBART takes as input the control cell state and the identity of the perturbed gene, and predicts the resulting transcriptional response across all genes. Specifically, for a perturbation targeting gene $g_{\text{pert}}$, the model receives:

- Gene IDs: $\mathbf{G} = [g_1, g_2, \ldots, g_N]$ for $N$ highly variable genes (HVGs)

- Unperturbed expression: $\boldsymbol{\mu} = [\mu_1, \mu_2, \ldots, \mu_N]$, where $\mu_i$ is the scVI denoised expression of gene $g_i$ in unperturbed cells

- Perturbation indicator: $\mathbf{p} = [p_1, p_2, \ldots, p_N]$, where

$$
p_i = \begin{cases} -10.0 & \text{if } g_i \text{ is inhibited (INH)} \\ 10.0 & \text{if } g_i \text{ is overexpressed (OE)} \\ 0.0 & \text{otherwise} \end{cases}
$$

The choice of ±10.0 was selected to provide a clear perturbation signal well-separated from the typical pretraining logFC distribution (median = -0.62, IQR = [-1.19, 1.15]), while remaining within the maximum observed range (±13). This magnitude ensures the model can distinguish fine-tuning perturbation indicators from observed logFC inputs during pretraining.

The model constructs input embeddings as in pretraining, except the logFC embedding component is initialized using the perturbation indicator:

$$
\mathbf{h}_i^{\text{ft}} = W_p[\mathbf{k}_i; W_{\text{logFC}} p_i; W_{\text{expr}} \log(1 + \mu_i)]
$$

where $\mathbf{k}_i = W_k[W_r \mathbf{r}_i; \mathbf{b}_i]$ combines pathway and BioBERT knowledge embeddings as in the pretraining architecture. These embeddings are processed through the encoder-decoder architecture. The encoder produces contextualized representations:

$$
\mathbf{z}_i = \text{Encoder}(\mathbf{h}_i^{\text{ft}}; \mathbf{h}_1^{\text{ft}}, \ldots, \mathbf{h}_N^{\text{ft}})
$$

which serve as memory for the decoder. The decoder outputs $\mathbf{d}_i$ are first passed through the embedding reconstruction head to obtain $\hat{\mathbf{h}}_i = W_{\text{rec2}} \cdot \text{GELU}(W_{\text{rec1}} \mathbf{d}_i)$, and the logFC prediction head then outputs $\hat{x}_i = W_{\text{out2}} \cdot \text{GELU}(W_{\text{out1}} \hat{\mathbf{h}}_i)$, where $W_{\text{out1}} \in \mathbb{R}^{512 \times 768}$ and $W_{\text{out2}} \in \mathbb{R}^{1 \times 512}$ are the same prediction head weights from pretraining.

### A.4.2. FINE-TUNING OBJECTIVE

Unlike pretraining, which uses a masked denoising objective, fine-tuning optimizes a multi-task loss combining weighted MSE, global correlation, and top-K correlation. Let $\hat{\mathbf{x}} = [\hat{x}_1, \ldots, \hat{x}_N]$ denote predicted logFC values and $\mathbf{x}^* = [x_1^*, \ldots, x_N^*]$ denote ground-truth logFC values computed from scVI differential expression analysis. The total loss is:

$$
\mathcal{L}_{\text{finetune}} = \alpha \cdot \mathcal{L}_{\text{wMSE}} + \gamma \cdot \mathcal{L}_{\text{corr}}
$$

where the weighted MSE loss places higher weight on genes with larger fold-changes:

$$
\mathcal{L}_{\text{wMSE}} = \frac{1}{N} \sum_{i=1}^{N} (1 + \lambda \cdot |x_i^*|) \cdot (\hat{x}_i - x_i^*)^2, \quad \lambda = 0.5
$$

The correlation loss combines global Pearson correlation across all genes with top-K correlation on the most differentially expressed genes:

$$
\mathcal{L}_{\text{corr}} = 1 - [\beta \cdot \rho_{\text{global}}(\hat{\mathbf{x}}, \mathbf{x}^*) + (1 - \beta) \cdot \rho_{\text{top}}(\hat{\mathbf{x}}, \mathbf{x}^*)]
$$

where $\rho_{\text{top}}$ is computed on the top 20% genes ranked by absolute true logFC ($|\mathbf{x}^*|$). The loss weights $\alpha$, $\beta$, and $\gamma$ are implemented as trainable scalar parameters (initialized to 1.0, 0.5, and 0.5, respectively) and constrained to the range $(0, 1)$ via sigmoid transforms, allowing the model to automatically balance the relative importance of magnitude accuracy versus ranking performance during training. These weights are jointly optimized with all model parameters via backpropagation using AdamW with learning rate $10^{-4}$. The Pearson correlation is computed in a differentiable manner to enable gradient flow:

$$
\rho_{\text{Pearson}}(\hat{\mathbf{x}}, \mathbf{x}^*) = \frac{\text{Cov}(\hat{\mathbf{x}}, \mathbf{x}^*)}{\sqrt{\text{Var}(\hat{\mathbf{x}}) + \epsilon} \cdot \sqrt{\text{Var}(\mathbf{x}^*) + \epsilon}}, \quad \epsilon = 10^{-8}
$$

Models were fine-tuned for up to 100 epochs with early stopping (patience=5).

### A.4.3. EVALUATION METRICS

We evaluated model performance using four complementary metrics that assess different aspects of perturbation prediction quality. For each metric, we identify the top-$K$ genes by absolute true logFC (ground truth ranking), where $K \in \{50, 100, 200, 300, 400, 500\}$, and then compute metrics on predicted logFC values for these true top-$K$ genes. This ground-truth-based evaluation focuses on the genes that are most significantly perturbed according to experimental evidence, ensuring that metrics directly assess how well the model captures the most important transcriptional changes. To reduce bias from dropout artifacts, we restricted evaluation to genes with non-zero proportions $\geq 0.3$ in both perturbed and control populations based on scVI posterior estimates.

**Cosine Similarity.** Cosine similarity measures the directional agreement between predicted and true logFC profiles for the top-$K$ genes. Let $\hat{\mathbf{x}}_K = [\hat{x}_{g_1}, \ldots, \hat{x}_{g_K}]$ and $\mathbf{x}_K^* = [x_{g_1}^*, \ldots, x_{g_K}^*]$ denote the predicted and true logFC vectors for the top-$K$ genes by ground truth. Cosine similarity is computed as:

$$\text{CosSim}(\hat{\mathbf{x}}_K, \mathbf{x}_K^*) = \frac{\hat{\mathbf{x}}_K \cdot \mathbf{x}_K^*}{\|\hat{\mathbf{x}}_K\|_2 \|\mathbf{x}_K^*\|_2} \tag{1}$$

where $\cdot$ denotes the dot product and $\| \cdot \|_2$ denotes the $L_2$ norm. Cosine similarity ranges from $-1$ to $1$, with higher values indicating better alignment of predicted expression changes with ground truth. This metric is scale-invariant and captures the overall pattern of gene regulation.

**Root Mean Squared Error (RMSE).** RMSE quantifies the magnitude of prediction errors for the top-$K$ genes identified by ground truth. RMSE is defined as:

$$\text{RMSE}(\hat{\mathbf{x}}_K, \mathbf{x}_K^*) = \sqrt{\frac{1}{K} \sum_{i=1}^{K} (\hat{x}_{g_i} - x_{g_i}^*)^2} \tag{2}$$

where $\hat{x}_{g_i}$ and $x_{g_i}^*$ are the predicted and true logFC values for gene $g_i$ in the top-$K$ set. Lower RMSE indicates more accurate predictions of expression change magnitudes. This metric is sensitive to outliers and penalizes large deviations more heavily than small ones.

**Enrichment Factor.** Enrichment factor (EF) measures the overlap between predicted and true top-$K$ DEGs relative to random expectation. Let $\mathcal{G}_{\text{pred}}$ denote the set of top-$K$ genes ranked by absolute predicted logFC $|\hat{x}_{g_i}|$, and $\mathcal{G}_{\text{true}}$ denote the set of top-$K$ genes ranked by absolute true logFC $|x_{g_i}^*|$. The enrichment factor is computed as:

$$\text{EF} = \frac{|\mathcal{G}_{\text{pred}} \cap \mathcal{G}_{\text{true}}| + \epsilon}{\frac{|\mathcal{G}_{\text{pred}}| \cdot |\mathcal{G}_{\text{true}}|}{|\mathcal{G}|} + \epsilon} \tag{3}$$

where $| \cdot |$ denotes set cardinality, $\mathcal{G}$ is the set of all genes in the evaluation space, and $\epsilon = 0.05$ is a pseudocount to prevent division by zero and stabilize the metric when the baseline proportion is very small. The denominator represents the expected overlap under random ranking. EF values greater than 1 indicate enrichment above random chance, with higher values reflecting more significant overlap and better recovery of truly differentially expressed genes.

**F1 Score for Direction Classification.** The F1 score assesses the accuracy of predicting gene response direction. For each gene $g_i$, we assign a direction label based on its logFC magnitude:

$$\text{label}(x_{g_i}^*) = \begin{cases} \text{up} & \text{if } x_{g_i}^* \geq \log_2(1.2) \\ \text{down} & \text{if } x_{g_i}^* \leq -\log_2(1.2) \\ \text{unchanged} & \text{otherwise} \end{cases} \tag{4}$$

where the threshold $\log_2(1.2) \approx 0.263$ corresponds to a 1.2-fold change. We compute precision, recall, and F1 score separately for upregulated and downregulated genes:

$$\text{Precision}_c = \frac{\text{TP}_c}{\text{TP}_c + \text{FP}_c + \epsilon}, \quad \text{Recall}_c = \frac{\text{TP}_c}{\text{TP}_c + \text{FN}_c + \epsilon} \tag{5}$$

$$\text{F1}_c = \frac{2 \cdot \text{Precision}_c \cdot \text{Recall}_c}{\text{Precision}_c + \text{Recall}_c + \epsilon} \tag{6}$$

where $c \in \{\text{up}, \text{down}\}$, $\text{TP}_c$, $\text{FP}_c$, and $\text{FN}_c$ denote true positives, false positives, and false negatives for class $c$ based on comparing $\text{label}(\hat{x}_{g_i})$ and $\text{label}(x^*_{g_i})$, and $\epsilon = 10^{-6}$ prevents division by zero. The final F1 score is reported as the mean of F1 scores for upregulated and downregulated classes:

$$\text{F1}_{\text{mean}} = \frac{\text{F1}_{\text{up}} + \text{F1}_{\text{down}}}{2} \tag{7}$$

This metric evaluates whether the model correctly identifies not only which genes respond to perturbation, but also the direction of their response.

### A.4.4. BASELINE MODELS

To benchmark the performance of scDEBART, we compared it against four baseline models: GEARS, scGPT, a multilayer perceptron (MLP), and linear regression (LR). Each model was trained under two input regimes: scVI denoised gene expression and sum-normalized counts.

**GEARS.** GEARS (Graph-Enhanced Gene Activation and Repression Simulator) is a graph neural network-based model that leverages gene regulatory networks to predict perturbation responses. The model constructs a gene interaction graph from existing knowledge bases (e.g., protein-protein interactions, co-expression networks) and uses Graph Attention Networks (GATs) to propagate perturbation signals through the regulatory network. We trained GEARS using the publicly available implementation from the official GitHub repository (`https://github.com/snap-stanford/GEARS`). Train/validation/test splits were defined using the random seeds provided by the GEARS framework to ensure reproducibility and fair comparison. Genes that could not be mapped to the pre-constructed gene regulatory graph were automatically excluded from the analysis, as GEARS requires network connectivity for message passing. For sum-normalized count data, we normalized raw counts by library size before feeding them into the model. For scVI-denoised expression data, we used the same paired perturbed and unperturbed samples as those used for scDEBART fine-tuning. The model was trained for 20 epochs with a hidden dimension of 64 and default hyperparameters (learning rate $10^{-3}$, weight decay $5 \times 10^{-4}$) as specified in the original GEARS publication.

**scGPT.** Unlike GEARS, scGPT requires explicitly paired control samples for each perturbed cell during training and inference. For scVI-denoised expression data, we used the same dataset as scDEBART fine-tuning, where scVI's stochastic sampling mechanism provides matched control samples from the learned posterior distribution for each perturbed cell. For sum-normalized counts, we randomly matched each perturbed cell with a control cell to create paired training instances. During inference, we generated 30 independent control pairings for each perturbed sample and aggregated predictions by computing the median across these replicates to obtain the final predicted expression. We followed the fine-tuning protocol from the Therapeutics Data Commons (TDC) (Velez-Arce et al., 2024) (`https://huggingface.co/tdc/scGPT`), training only the gene encoder, value encoder, and expression decoder modules for 10 epochs with MSE loss.

**MLP baseline.** The MLP baseline takes as input the concatenation of a pre-trained gene embedding (BioBERT + MsigDB C2 pathway features) for the perturbed gene and the unperturbed expression vector of the top 5,000 highly variable genes (HVGs). For scVI-denoised inputs (scVI-MLP), we follow the same pairing strategy as other scVI-based models: each perturbed sample is matched to an unperturbed scVI-denoised control profile, and the concatenated control expression and gene embedding are used to predict the perturbation-induced log fold-change over HVGs, using the scVI-based DE logFC as the training target. For sum-normalized counts (counts-MLP), we construct a separate dataset in which each perturbed sample is randomly paired with an unperturbed control cell in raw counts; the input again concatenates the control expression vector and the gene embedding, but the model is trained to predict perturbed absolute expression. At evaluation time, both predictions and ground-truth expressions are converted to log fold-changes using the matched control expression, $\log_2\{(\hat{x} + \epsilon)/(x_{\text{ctrl}} + \epsilon)\}$ with $\epsilon = 10^{-8}$, where $\hat{x}$ denotes the predicted perturbed expression and $x_{\text{ctrl}}$ the corresponding control expression for each gene and perturbation. The MLP consists of three hidden layers with widths 4,096, 2,048, and 1,024, GELU activations, and dropout rate 0.1. Models are trained with AdamW (learning rate $10^{-3}$, weight decay $10^{-4}$), batch size 2,048 (in the main experiments), for up to 200 epochs with early stopping (patience = 5), using MSE loss on the HVG outputs. All perturbed–control pairings and train/validation/test splits are identical to those used for scDEBART and scGPT in both the scVI-denoised and count-based settings.

**Linear regression baseline.** The linear regression (LR) baseline uses the same input–output formulation as the MLP baseline but replaces the nonlinear network with a single fully connected layer. For scVI-denoised inputs (scVI-LR), the model takes as input the concatenation of the gene embedding and the matched unperturbed scVI-denoised HVG expression vector, and directly predicts log fold-change over HVGs, trained with MSE loss on the scVI-based DE logFC targets. The

scVI-LR model is optimized with AdamW (learning rate $10^{-3}$, weight decay $10^{-4}$), using a batch size of 4,096, up to 200 epochs with early stopping (patience = 5, $\Delta_{\min} = 0$). For sum-normalized counts (counts-LR), the same concatenated input (gene embedding plus randomly matched unperturbed count vector) is used to predict perturbed absolute expression; these predictions are then converted to log fold-changes with respect to the paired control counts via $\log_2\{(\hat{x} + \epsilon)/(x_{\mathrm{ctrl}} + \epsilon)\}$ with $\epsilon = 10^{-8}$, analogous to the counts-MLP setting. The counts-LR model is trained with AdamW (learning rate $10^{-3}$, weight decay $10^{-4}$), batch size 2,048, and early stopping over a maximum of 300 epochs (patience = 5). As with the MLP baseline, all perturbed–control pairs and data splits for LR exactly match those used for scDEBART and scGPT.

**Conversion from absolute expression to log fold-change.** For scVI-MLP and scVI-LR, the targets and outputs were logFC by construction, so no additional transformation was required before evaluation. In contrast, GEARS, scGPT, the raw-count MLP, and the raw-count LR produced absolute expression predictions. For these models, we converted predictions and corresponding ground truths into log fold-changes using their paired or averaged control expression profiles: for each gene, we computed $\log_2\{(\hat{x} + 10^{-8})/(x_{\mathrm{ctrl}} + 10^{-8})\}$, where $\hat{x}$ is the predicted perturbed expression and $x_{\mathrm{ctrl}}$ is the matched control expression (GEARS: dataset-wide control mean; scGPT/MLP/LR: paired control expression). This ensured that all methods were evaluated on a common logFC scale consistent with the DE targets used by scDEBART.

## A.5. Cross-Modal Transfer on Drug Perturbations

### A.5.1. DRUGBANK DRUG-TARGET INTERACTION EXTRACTION

We extracted drug-target interactions (DTIs) from DrugBank v5.1.13 (Knox et al., 2024) to map SCIPLEX compounds to their primary protein targets within the scDEBART gene vocabulary. We retained only primary targets annotated with action types "inhibitor" or "antagonist" to align with the CRISPRi knockdown mechanism in Replogle K562. Off-target interactions and activator/agonist annotations were excluded. This filtering yielded 220 DTIs covering 65 compounds, with a median of 2 target genes per drug (range: 1–20). The distribution of target gene counts per compound is shown in Appendix Figure 12.

### A.5.2. SCIPLEX DATA PREPROCESSING AND DIFFERENTIAL EXPRESSION ANALYSIS

We used the publicly available SCIPLEX dataset (Srivatsan et al., 2020), retaining only K562 cells to match the cell line used in Replogle K562 fine-tuning. Quality control followed CELLxGENE corpus standards: cells with total UMI counts or mitochondrial read fractions beyond $\pm 3$ standard deviations were removed, yielding 158,293 cells. Genes were filtered to protein-coding genes in the scDEBART vocabulary (19,742 genes).

SCIPLEX includes four drug concentrations (10, 100, 1000, 10000 nM). For all doses combined, we trained a single scVI model with 64 latent dimensions, using drug identity as batch key (200 max epochs, early stopping patience 5). For each compound-dose combination, we computed pairwise differential expression between drug-treated and vehicle control cells using scVI's posterior sampling framework, following the same protocol as Perturb-seq datasets (Appendix A.2.3), including logFC calculation, Proba-DE estimation, and non-zero proportion filtering.

### A.5.3. EVALUATION PROTOCOL

For each compound with DrugBank-derived target genes, we simulated multi-gene perturbation by setting perturbation indicators $p_i = -10.0$ for all target genes and $p_i = 0$ otherwise, matching the CRISPRi mechanism and Replogle K562 fine-tuning protocol. For compounds with multiple targets (Appendix Figure 12), all targets were perturbed simultaneously as a multi-gene knockout. The model predicted transcriptome-wide logFC, which was evaluated using the same metrics and protocols as genetic perturbations (Appendix A.4): cosine similarity, RMSE, enrichment factor, and F1 score on top-20 DEGs. Across all genes, logFC distributions were tightly centered near zero (e.g., 10 nM: IQR [-0.21, 0.22]; 10000 nM: IQR [-0.44, 0.27]), with occasional outliers (min/max at 10000 nM: -2.46/2.17). The limited logFC magnitudes necessitated restricting evaluation to top-20 DEGs, as even these genes showed modest effect sizes.

Evaluation was performed using the Replogle K562 fine-tuned scDEBART model with pretrained under three random seeds (30, 40, 50) across four doses (10, 100, 1000, 10000 nM). Baseline models (scVI-scGPT) followed identical preprocessing, multi-gene perturbation simulation, and evaluation procedures. For scGPT, predicted absolute expression was converted to logFC by normalizing against vehicle control predictions, consistent with genetic perturbation evaluation (Appendix A.4.3).

## A.6. Additional Experimental Results

### A.6.1. GENE DETECTION ANALYSIS OF PREDICTED DEGS

Baseline models exhibit relatively high correlation but lower enrichment factors compared to scDEBART, suggesting that their top predictions may include false positives. To investigate characteristics of predicted DEGs, we examined their detection rates (non-zero proportions).

We computed the median non-zero proportion (across perturbed and unperturbed cells) for the top-50 predicted DEGs of each test perturbation across five Perturb-seq datasets. scDEBART selected genes with median non-zero proportions of 0.60 (perturbed condition) and 0.73 (unperturbed condition), indicating that the model prioritizes genes with reliably measurable expression in both states. In contrast, baseline models showed substantially lower detection rates: scVI-scGPT (0.13 perturbed, 0.10 unperturbed), counts-scGPT (0.23, 0.22), scVI-GEARS (0.28, 0.19), counts-GEARS (0.13, 0.09), and counts-MLP (0.12, 0.11) (Appendix Figure 13). These values frequently fall below the 0.3 threshold used to filter lowly expressed genes during pretraining corpus construction, suggesting that baseline models frequently rank genes with high dropout rates among their top predictions. The association between low detection rates in baseline predictions and their reduced enrichment factors suggests that dropout-prone genes may contribute to false positive predictions.

scVI-MLP and scVI-LR exhibited non-zero proportions of 1.0 in both conditions, which initially appears optimal. However, manual inspection revealed that these models consistently predicted the same small subset of genes with large absolute logFC values regardless of the perturbed gene, resulting in nearly identical top-50 gene sets across all test perturbations. We followed standard practices for MLP/LR training (Appendix A.4.4), but these models struggled to learn perturbation-specific patterns from scVI-denoised expression without large-scale pretraining. This perturbation-independent behavior indicates that the models learned a fixed gene signature rather than perturbation-specific transcriptional responses.

This analysis reveals two distinct patterns in baseline model predictions: ranking genes with low detection rates (scGPT and GEARS variants, counts-MLP) or producing perturbation-independent gene sets (scVI-MLP, scVI-LR). Both patterns are associated with reduced enrichment factors, though the relative contribution of each factor remains unclear.

### A.6.2. COMPARISON WITH DESEQ2 PSEUDOBULK DE ANALYSIS

Throughout this work, we used scVI-based differential expression for training, validation, and testing, as it provides GPU-accelerated inference and explicitly models dropout and batch effects in single-cell RNA-seq data, making it well suited for large-scale perturbation modeling. In parallel, pseudobulk aggregation followed by DESeq2 remains a widely used strategy for single-cell DE analysis. However, pseudobulk + DESeq2 requires fitting a separate model for each comparison, so the computational cost grows with the number of perturbation–control pairs, whereas scVI amortizes this cost by learning a dataset-level generative model from which pairwise DE can then be computed efficiently for many comparisons. To assess whether scDEBART's predicted expression changes, trained against scVI-based DE, generalize to an alternative DE framework, we constructed a DESeq2 pseudobulk benchmark and evaluated all models against this alternative ground truth.

**Pseudobulk construction and DESeq2 analysis.**  For each perturbation (including the non-targeting control), we grouped cells by perturbation and randomly partitioned them into up to three pseudobulk replicates. Within each replicate, we summed raw UMI counts across cells to obtain a gene-by-replicate count vector, yielding a pseudobulk count matrix whose rows correspond to perturbation–replicate samples and whose columns correspond to genes.

For each perturbation, we then performed a DESeq2 analysis comparing pseudobulk samples from that perturbation against pseudobulk samples from non-targeting controls. Low-count genes with total counts $< 10$ across all samples were removed prior to model fitting, and DESeq2 was run with default settings.

To rank genes by differential expression, we used DESeq2's Wald test statistic (`stat`), which integrates both the magnitude of log-fold change and its statistical significance into a single ranking criterion (`log2FoldChange` divided by `lfcSE`) (Love et al., 2014). Unlike log-fold change alone, the test statistic accounts for estimation uncertainty and is commonly used to select top differentially expressed genes in a principled manner. Because the test statistic does not directly represent fold-change magnitude, we focused our evaluation on ranking-based metrics—cosine similarity and enrichment factor—rather than absolute deviation metrics such as RMSE. Similarly, we did not compute F1 score, as establishing a universal threshold for the test statistic (analogous to a fold-change cutoff of 1.2) is less intuitive and dataset-dependent. These two metrics are sufficient to assess whether predicted gene rankings align with pseudobulk-based DE quantification.

**scDEBART's performance on DESeq2 pseudobulk DE.** We evaluated all models across five Perturb-seq datasets (Norman K562, Replogle K562, Replogle RPE1, Nadig Jurkat, and Nadig HepG2), computing cosine similarity and enrichment factor at multiple top-$k$ cutoffs (50, 100, 200, 300, 400, and 500 DEGs) over three random seeds (Figure 14).

For top-50 genes, scDEBART achieved mixed performance across datasets. In cosine similarity, scDEBART ranked first on two datasets: Nadig HepG2 (0.613 vs scVI-scGPT's 0.608, $+0.005$) and Nadig Jurkat (0.518 vs scVI-scGPT's 0.482, $+0.036$). On Replogle RPE1 and Replogle K562, scDEBART ranked second (0.652 vs scVI-scGPT's 0.662, $-0.010$; and 0.533 vs counts-MLP's 0.562, $-0.029$, respectively). However, on Norman K562, scDEBART showed notably weaker performance, ranking sixth with cosine similarity of 0.365 compared to scVI-GEARS's 0.483 ($-0.118$). This dataset-dependent variation suggests that cosine similarity on pseudobulk DE is sensitive to dataset-specific characteristics such as cell type composition, sequencing depth, and perturbation effect sizes.

In enrichment factor, scDEBART demonstrated more consistent performance, ranking first on four of five datasets: Nadig HepG2 (8.48 vs scVI-GEARS's 6.55, $+1.93$), Nadig Jurkat (5.07 vs scVI-GEARS's 4.41, $+0.66$), Replogle RPE1 (9.79 vs scVI-GEARS's 6.84, $+2.95$), and Norman K562 (3.49 vs scVI-scGPT's 2.05, $+1.44$). On Replogle K562, scVI-GEARS ranked first (6.89 vs scDEBART's 5.10, $-1.79$), with scDEBART ranking third. scDEBART substantially outperformed all counts-based variants across all five datasets, indicating robust DEG prioritization despite methodological differences in DE quantification.

Across the ten comparisons (five datasets × two metrics), scDEBART achieved first-place rankings in six instances (two cosine similarity, four enrichment factor), more than any other baseline. This result indicates that scDEBART maintains robust performance across DE methodologies, even when evaluated against an alternative framework distinct from its training procedure.

## A.7. Ablation Studies

### A.7.1. IMPACT OF PRETRAINING AND CELL FILTERING ON PREDICTION PERFORMANCE

To isolate the contributions of two key design choices—large-scale pretraining and cell-level quality filtering—we conducted ablation studies on the Norman K562 dataset. We compared three configurations: (1) **scDEBART (full model)**, the complete pipeline with both pretraining and filtering, (2) **without pretrain**, training solely on the Norman K562 dataset without pretraining, and (3) **without filtering**, where all cells were used for DE computation without applying perturbation-response-based quality control (Appendix A.2.2). All models were evaluated on their respective datasets (filtered or unfiltered) using the same 80/10/10 train/val/test split and three random seeds.

Appendix Figure 15 shows performance across top-$K$ genes ($K \in \{50, 100, 200, 300, 400, 500\}$) for four metrics, with top-50 results summarized in Appendix Table 3. All values are reported as median $\pm$ standard deviation across three random seeds. The full model achieved cosine similarity of $0.852 \pm 0.173$, enrichment factor of $14.901 \pm 4.919$, and F1 score of $0.684 \pm 0.182$ at top-50.

**Impact of Pretraining.** Removing pretraining degraded performance across most metrics. At top-50, enrichment factor decreased from $14.901 \pm 4.919$ to $12.693 \pm 4.865$ (14.8% decrease, ks-test $p = 6.58 \times 10^{-7}$), F1 score dropped from $0.684 \pm 0.182$ to $0.616 \pm 0.187$ (9.9% decrease, $p = 1.48 \times 10^{-3}$), cosine similarity decreased to $0.825 \pm 0.185$ (3.2% decrease, $p = 0.251$), and RMSE increased to $0.523 \pm 0.208$ (5.9% increase, $p = 0.181$) (Appendix Figure 15). The pretraining effect is particularly pronounced for cosine similarity at larger $K$, with the gap between the full and pretraining-ablated models widening as $K$ increases, indicating that pretraining is critical for maintaining accurate gene ranking across the full predicted ranking, not only for identifying top differentially expressed genes.

**Impact of Cell Filtering.** Removing cell-level quality filtering in Perturb-seq data yielded modest effects: at top-50, cosine similarity decreased by 2.7% to $0.829 \pm 0.168$ ($p = 0.035$), enrichment factor decreased by 3.7% to $14.349 \pm 4.663$ ($p = 0.096$), F1 score decreased by 3.7% to $0.659 \pm 0.220$ ($p = 0.013$), while RMSE improved by 6.3% to $0.463 \pm 0.284$ ($p = 0.541$). Most metrics showed modest changes, suggesting that pretraining dominates the contribution. However, cells perturbed by the same gene exhibit substantial heterogeneity in expression space (Appendix Figure 11). Following Song et al., we applied stringent filtering that retained only 31–50% of labeled cells to enrich for strong transcriptional responses. Despite this reduction in training volume, performance remained comparable to the full dataset, demonstrating that improved data quality can offset reduced sample size. We therefore retain filtering to prioritize signal quality over sample quantity and minimize confounding from weakly perturbed cells.

| Model | Cosine similarity | RMSE | EF | F1 |
|---|---|---|---|---|
| scDEBART (full) | $0.852 \pm 0.173$ | $0.494 \pm 0.208$ | $14.901 \pm 4.919$ | $0.684 \pm 0.182$ |
| without pretrain | $0.825 \pm 0.185$ | $0.523 \pm 0.208$ | $12.693 \pm 4.865$ | $0.616 \pm 0.187$ |
| without filtering | $0.829 \pm 0.168$ | $0.463 \pm 0.284$ | $14.349 \pm 4.663$ | $0.659 \pm 0.220$ |

*Table 3.* **Ablation study results on top-50 DEGs.** Metrics computed on top-50 genes in the Norman K562 test set, shown as median $\pm$ standard deviation over three random seeds. F1 score uses a 1.2-fold change threshold.

### A.8. Effect of Pretraining Corpus Size

To assess whether downstream performance scales with the size of the pretraining corpus, we conducted scaling experiments using 25%, 50%, and 100% of the 6.28M DE profiles. All configurations used identical model architecture and hyperparameters; only the number of pretraining profiles was varied.

**Pretraining reconstruction loss.** Pretraining validation MSE decreased consistently with corpus size: 0.0275 (25%), 0.0268 (50%), and 0.0182 (100%). Notably, the full corpus (100%) not only achieves the lowest final MSE but also converges faster, reaching its plateau around epoch 20 compared to epoch 25 for 50% and epoch 40 for 25%, indicating that larger corpora yield both faster and better reconstruction of logFC targets. Figure 16 shows validation loss curves across all three corpus sizes.

**Downstream perturbation prediction.** We evaluated downstream performance on Norman K562 for each corpus size subset; results are shown in Figure 17. At top-50 DEGs, EF increased from 12.88 (25%) and 12.51 (50%) to 14.33 at full corpus (100%), and F1 improved from 0.576 and 0.573 to 0.609. Cosine similarity remained relatively stable across corpus sizes (0.772, 0.779, and 0.775), while RMSE decreased monotonically. This pattern is consistent across top-100 and top-200 DEG thresholds, with EF and F1 showing a clear upward step at 100% in contrast to the plateau between 25% and 50%. Together, these results indicate that the full pretraining corpus provides a meaningful benefit over subsets, with gains primarily emerging from the 50% to 100% transition rather than accumulating gradually across corpus sizes.

### A.9. Effect of Supervision Quality: scVI Target Noising

The scaling experiments in Appendix A.8 hold the pretraining objective constant across corpus sizes, and therefore cannot isolate the contribution of supervision quality from that of data scale. A complementary question is whether the quality of scVI-derived logFC targets—rather than their quantity—independently contributes to downstream performance. If downstream performance benefits stem in part from the reliability of scVI-denoised supervision, then deliberately degrading target quality should lead to measurable performance degradation. To test this hypothesis, we performed a target-noising experiment in which uncertainty-proportional Gaussian noise was injected into the scVI-derived logFC training targets during fine-tuning on the Norman K562 dataset. The key design choice is that noise magnitude is calibrated to the gene-wise posterior uncertainty estimated by scVI itself: genes whose logFC estimates are less certain receive proportionally larger noise, making this a principled probe of supervision reliability rather than an arbitrary perturbation.

**Experimental setup.** For each gene $g$ in a training perturbation profile, we generated noisy logFC targets by adding gene-wise Gaussian noise proportional to the scVI-estimated logFC uncertainty (posterior standard deviation $\sigma_g$):

$$\tilde{x}_g = x_g + \gamma \cdot \sigma_g \cdot \epsilon, \quad \epsilon \sim \mathcal{N}(0, 1)$$

where $\gamma \in \{0.5, 1.0\}$ controls the noise magnitude, and $\tilde{x}_g$ is subsequently clipped to the scVI posterior range $[\min_g, \max_g]$. The noisy targets $\tilde{x}_g$ were used as training supervision during fine-tuning, while evaluation was performed against the original clean posterior median logFC as ground truth. Each configuration was run with a single random seed; the baseline uses clean targets ($\gamma = 0$).

**Results.** Table 4 reports performance on top-50 DEGs across four metrics. All metrics degraded monotonically as $\gamma$ increased: cosine similarity decreased from 0.832 ($\gamma = 0$) to 0.829 ($\gamma = 0.5$) and 0.827 ($\gamma = 1.0$); EF from 15.992 to 15.453 and 15.370; and F1 from 0.693 to 0.677 and 0.668. RMSE increased from 0.572 to 0.573 and 0.584, consistent with noisier supervision producing less accurate magnitude predictions.

*Table 4.* Effect of injecting uncertainty-proportional Gaussian noise into scVI-derived logFC training targets on Norman K562, evaluated on top-50 DEGs against clean ground truth. Mean ± std over test perturbations.

| Training targets | Cosine similarity ↑ | RMSE ↓ | EF ↑ | F1 ↑ |
|---|---|---|---|---|
| Clean ($\gamma = 0$) | $0.832 \pm 0.114$ | $0.572 \pm 0.197$ | $15.992 \pm 4.597$ | $0.693 \pm 0.146$ |
| Noisy ($\gamma = 0.5$) | $0.829 \pm 0.114$ | $0.573 \pm 0.190$ | $15.453 \pm 4.408$ | $0.677 \pm 0.145$ |
| Noisy ($\gamma = 1.0$) | $0.827 \pm 0.112$ | $0.584 \pm 0.190$ | $15.370 \pm 4.326$ | $0.668 \pm 0.147$ |

**Interpretation.** The monotonic degradation across all metrics is consistent with supervision quality contributing to downstream performance beyond data scale alone. However, several limitations apply. First, noising was applied only at the fine-tuning stage; a more complete test would require noising the pretraining corpus itself. Second, the modest degradation magnitude may reflect the limited scope of fine-tuning-stage noise relative to the full pretraining signal—the model may already have learned robust representations during pretraining that partially compensate for noisier fine-tuning targets. Accordingly, this experiment provides suggestive but not definitive evidence; a definitive decomposition of supervision quality versus data scale remains an important direction for future work.

## A.10. Computational Requirements

**Pretraining:** Pretraining scDEBART on 6.28M differential expression profiles required approximately 113 hours of wall-clock time (approximately 452 GPU-hours) on 4 × NVIDIA H100 GPUs (80GB memory) using mixed precision (bfloat16) and DeepSpeed ZeRO-3 optimization. Training was parallelized across 4 GPUs and converged at epoch 22 of 50.

**Fine-tuning:** Fine-tuning on individual Perturb-seq datasets required 1–5 GPU-hours per epoch on a single H100 GPU, totaling 10–50 GPU-hours per dataset depending on dataset size. All experiments converged within 5–10 epochs using early stopping with patience of 5 epochs.

## A.11. Data Availability

All datasets used in this study are publicly available:

- **CELLxGENE corpus:** Downloaded from `https://cellxgene.cziscience.com/` (accessed January 30, 2025)

- **Norman K562:** Norman et al. (2019), GEO accession GSE90546

- **Replogle K562/RPE1:** Replogle et al. (2022), available at `https://gwps.wi.mit.edu/`

- **Nadig HepG2/Jurkat:** Nadig et al. (2025), GEO accession GSE264667

- **DrugBANK:** Knox et al. (2024), available at `https://go.drugbank.com/releases/latest`

- **SCIPLEX:** Srivatsan et al. (2020), GEO accession GSE139944

BioBERT embeddings were generated using the publicly available BioBERT-Base v1.1 model (`dmis-lab/biobert-base-cased-v1.1`) from HuggingFace. Pathway annotations were obtained from MSigDB v2024.1 (`https://www.gsea-msigdb.org/`).

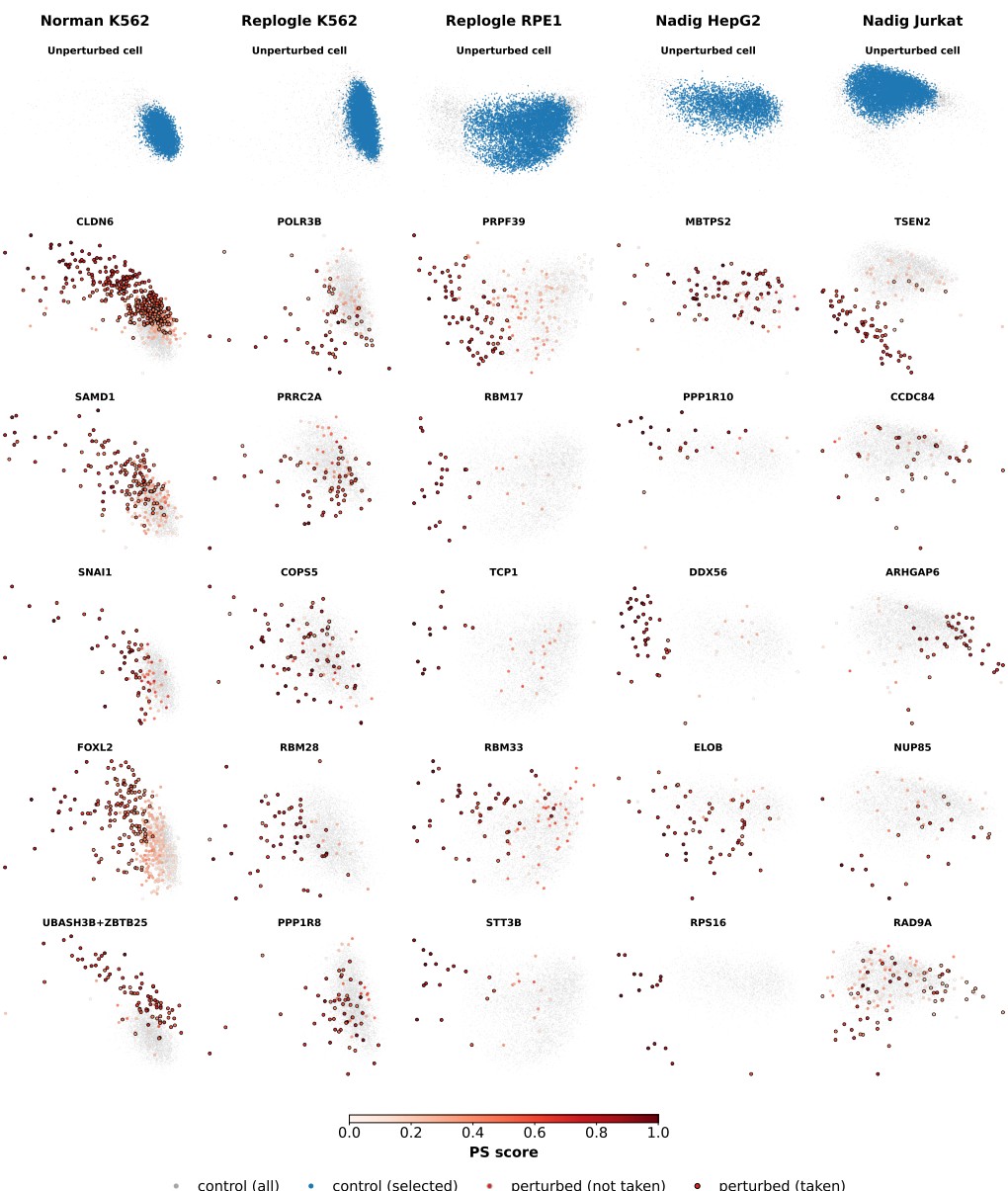

*Figure 11.* **Effect of PS-based filtering on perturbed cells in PCA space.** Each column corresponds to one Perturb-seq dataset. Top row: PCA embedding of selected control cells (blue) among all control cells (grey). Subsequent rows: For five randomly selected perturbations per dataset, control cells (grey) with perturbed cells colored by PS (color bar) and closely-formed final selected cells are black-bordered.

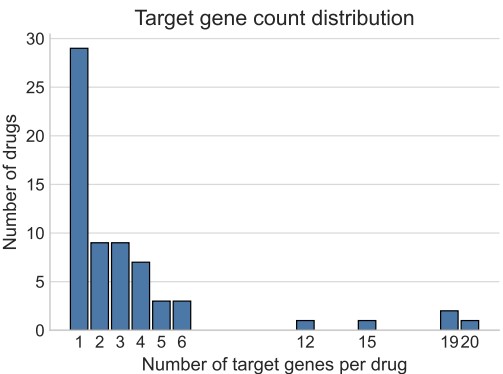

*Figure 12.* **Distribution of primary target gene counts per drug.** Histogram showing the number of SCIPLEX compounds annotated with varying numbers of primary target genes in DrugBank.

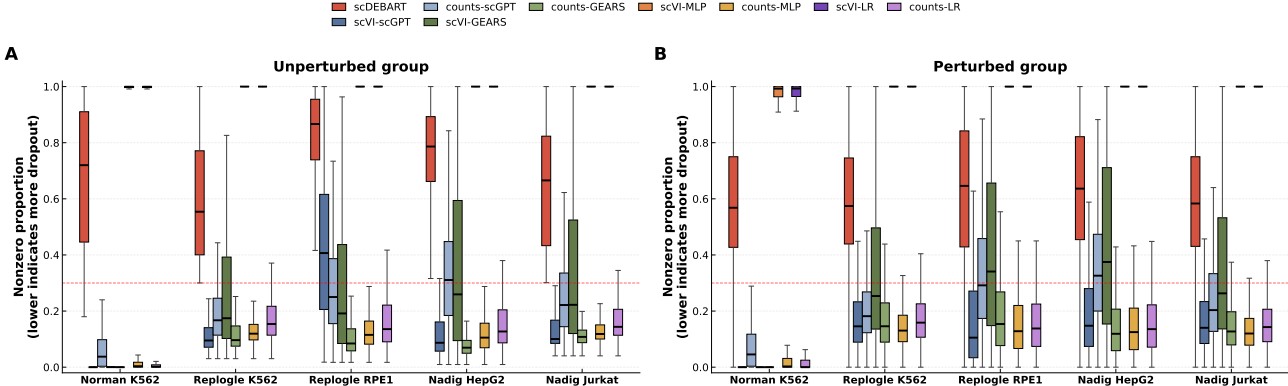

*Figure 13.* **Dropout sensitivity analysis of predicted top-50 DEGs.** Non-zero proportions of predicted top-50 genes in (A) unperturbed and (B) perturbed cell populations. scDEBART consistently predicts genes with high detection rates (median > 0.6), whereas baseline models frequently select genes with low detection rates, suggesting false positives driven by technical dropout rather than biological signal.

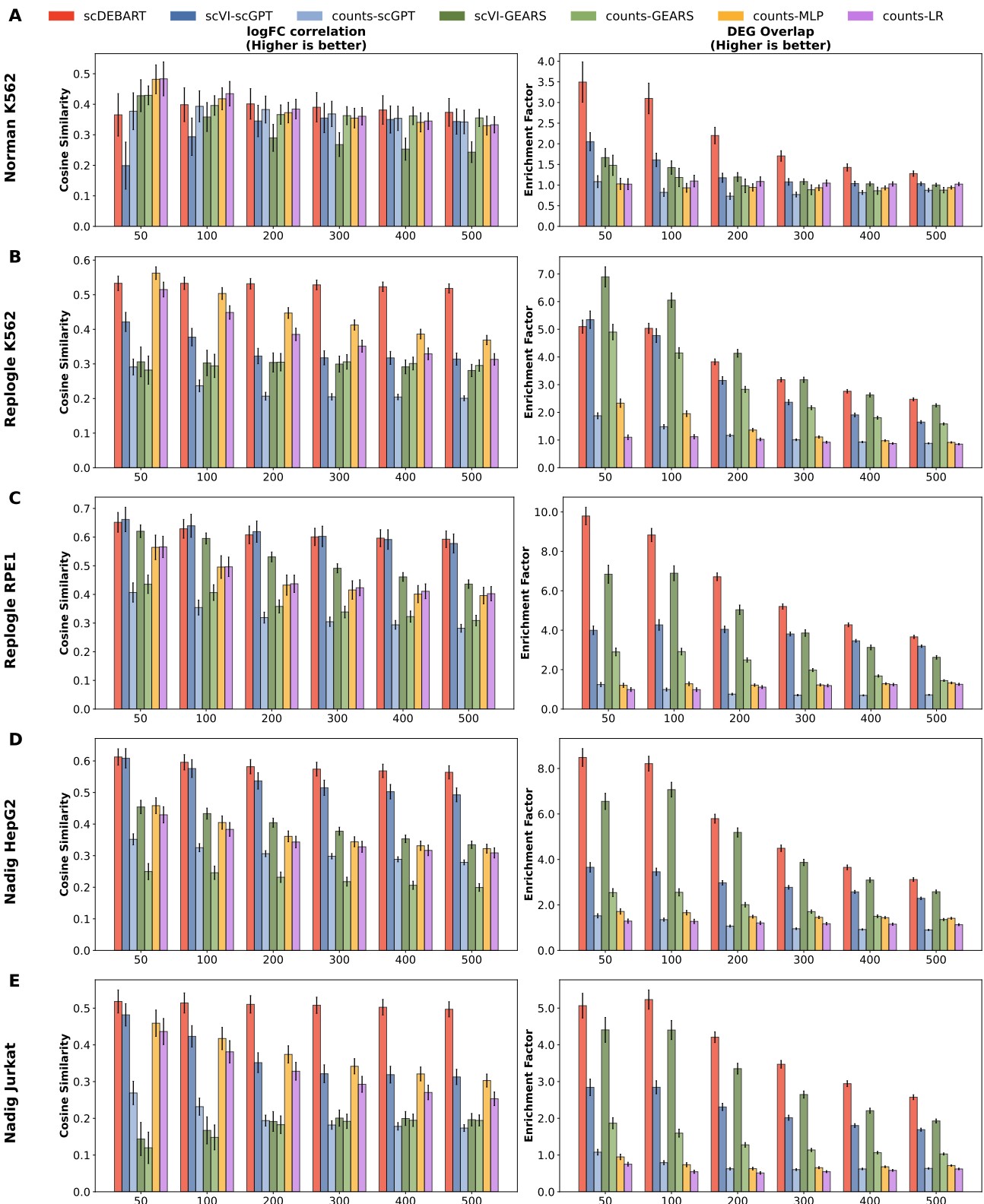

*Figure 14.* **Performance on DESeq2 pseudobulk DE across five Perturb-seq datasets.** Cosine similarity (left) and enrichment factor (right) are shown for multiple top-$k$ cutoffs (50, 100, 200, 300, 400, 500) across five datasets: (A) Norman K562, (B) Replogle K562, (C) Replogle RPE1, (D) Nadig HepG2, and (E) Nadig Jurkat. Metrics are aggregated over three random seeds. scVI-MLP and scVI-LR are excluded due to perturbation-independent prediction behavior (see Appendix A.6.1).

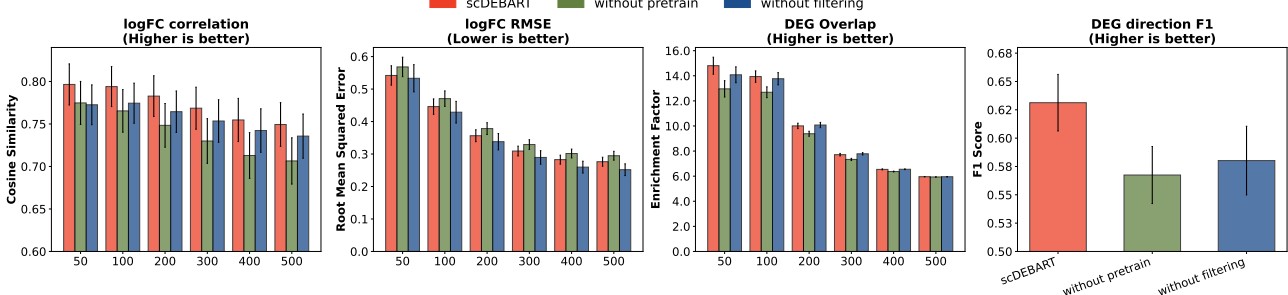

*Figure 15.* **Ablation study across top-$K$ DEGs.** Performance of the full scDEBART model (red) compared to ablated variants: without pretraining (green) and without cell filtering (blue). Four metrics are shown across top-$K$ genes ($K \in \{50, 100, 200, 300, 400, 500\}$): (A) logFC cosine similarity, (B) RMSE, (C) enrichment factor, and (D) F1 score (computed at top-50 only, using 1.2-fold threshold). Removing pretraining substantially reduces enrichment factor (14.8% at top-50, $p = 6.58 \times 10^{-7}$) and F1 score (9.9%), while removing filtering has more modest effects (3.7% reduction in EF, $p = 0.096$). Error bars represent 95% confidence intervals over three random seeds.

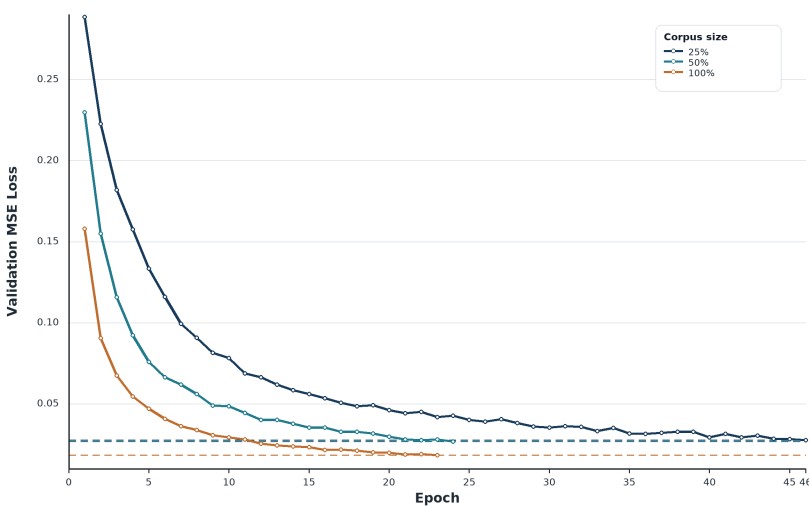

*Figure 16.* Pretraining validation MSE loss as a function of training epoch across three corpus sizes (25%, 50%, and 100% of the 6.28M DE profiles). Each curve shows per-epoch validation MSE on a held-out set of DE profiles, with the dashed horizontal line indicating the final converged loss for each configuration.

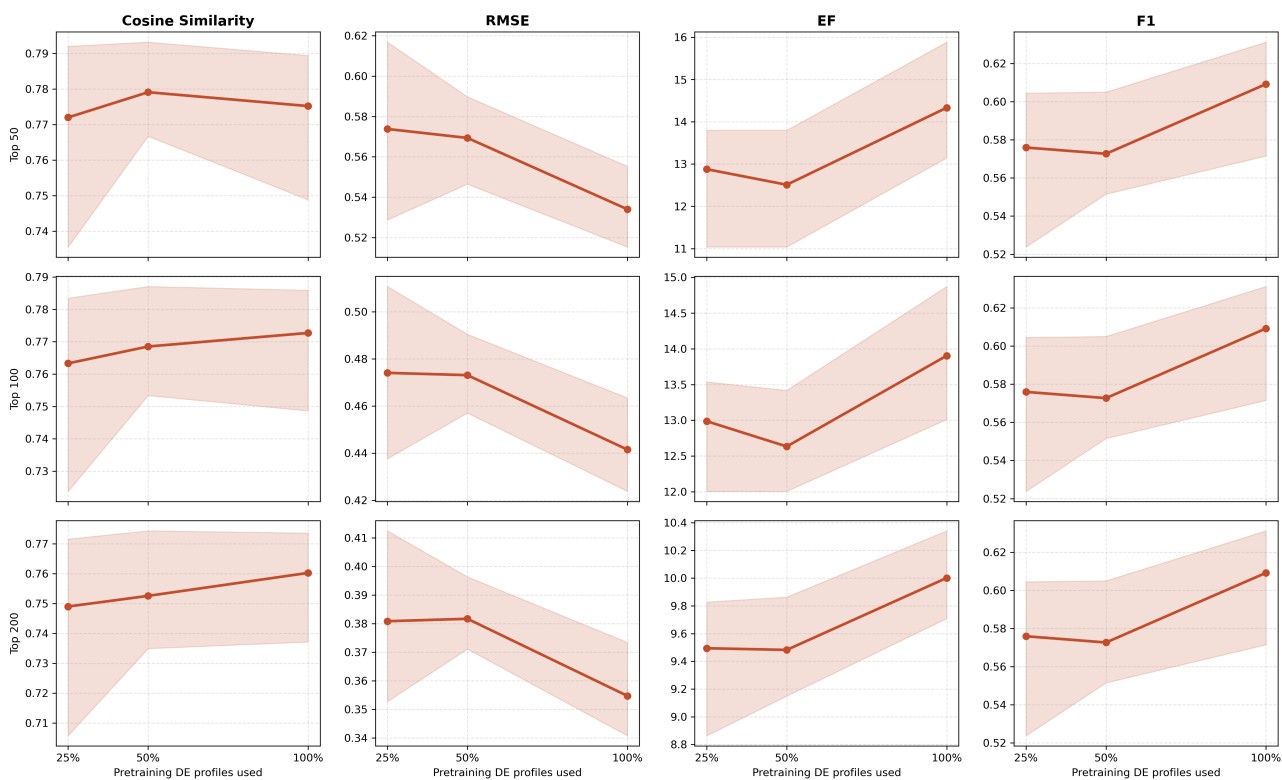

*Figure 17.* Downstream perturbation prediction performance on Norman K562 as a function of pretraining corpus size (25%, 50%, 100% of 6.28M DE profiles), evaluated across three top-$K$ DEG thresholds (Top 50, 100, 200). Each panel shows mean performance over 3 random seeds; shaded bands indicate the range across seeds. Columns correspond to four metrics: cosine similarity (higher is better), RMSE (lower is better), enrichment factor (EF; higher is better), and F1 score (higher is better).

