# OpenReview forum: "scDEBART: Predicting in silico Single-Cell Perturbation Responses via Large-Scale Differential Expression Learning"
_ICML.cc/2026/Conference — ICML 2026 regular_

### Official Review · Reviewer_1JZC · 2026-03-12

**Soundness:** 2
**Presentation:** 3
**Significance:** 3
**Originality:** 3
**Overall Recommendation:** 4
**Confidence:** 4

**Summary:**

In this study, the authors introduce scDEBART, a foundation model pretrained to predict log fold-changes (logFC) conditioned
on basal expression, thereby learning how gene sets co-vary across basal states at scale. The comparison results showed scDEBART achieved the best performance compared with based models.

**Compliance With Llm Reviewing Policy:**

Affirmed.

**Key Questions For Authors:**

A major comment is: To justify or compare the new moels and datasets like the popular large perturbation datasets like scPerturb and Tahoe-100m, as well as a set of models of modeling perturbation (like the diffusion models), to prove the soundness of the proposed model.

**Limitations:**

The validation might not be solid. There are popular large perturbation datasets like scPerturb and Tahoe-100m, as well as a set of models of modeling perturbation (like the diffusion models), which are not used for validation.

**Strengths And Weaknesses:**

Strengths: A novel model, with the best performance on some datasets with some base models.


Weaknesses:
There are popular large perturbation datasets like scPerturb and Tahoe-100m, as well as a set of models of modeling perturbation (like the diffusion models), which are not used for validation.

---

> ### Author Rebuttal · Authors · 2026-03-31
>
> Dear Reviewer 1JZC,
>
> Thank you for emphasizing the importance of broader benchmarking. Before addressing the specific resources you mention, we note that the current submission already evaluates the method in the main CRISPR-based perturbation-prediction setting across five Perturb-seq datasets from three independent studies, compares against GEARS, scGPT, MLP, and LR under matched count-based and scVI-denoised preprocessing, and includes both an orthogonal DESeq2 pseudobulk benchmark and an exploratory cross-modal transfer test on SCIPLEX.
>
> **Regarding scPerturb.** This resource spans multiple perturbation types and molecular readouts. As the scPerturb paper notes, "joint analysis is limited by the complexity of data integration" and "different experimental designs result in various covariates that may need to be accounted for" (Peidli et al., 2024). We therefore evaluated the constituent CRISPR Perturb-seq datasets directly, as these are most closely matched to our task—and notably, Norman K562 and Replogle K562/RPE1 used in our evaluation are themselves core components of scPerturb's CRISPR subset. Our benchmark additionally includes Nadig HepG2 and Jurkat, datasets published after scPerturb's release, extending coverage to more recent large-scale perturbation studies.
>
> **Regarding Tahoe-100M.** The primary task in this submission is genetic perturbation prediction, and drug perturbation is treated as an exploratory out-of-domain transfer setting rather than a co-equal benchmark. We used SCIPLEX for this purpose and are open to extending to Tahoe-100M in future work.
>
> **Regarding model comparisons.** Our baselines were selected to cover network-based (GEARS), foundation model (scGPT), and simple regression (LR/MLP) families. We agree that diffusion-based perturbation models represent a compelling and increasingly important direction for this task. However, a fair comparison would require re-running them under the same preprocessing pipeline, gene vocabulary, train/test splits, and logFC-based evaluation used in our study. We were not able to complete such a fully matched comparison within the rebuttal window, and we will state this limitation more clearly in the revision.

---

> > ### Author Rebuttal · Reviewer_1JZC · 2026-04-02
> >
> > The comments are answered.

---

### Official Review · Reviewer_5iuK · 2026-03-12

**Soundness:** 3
**Presentation:** 3
**Significance:** 3
**Originality:** 3
**Overall Recommendation:** 5
**Confidence:** 4

**Summary:**

This paper proposes scDEBART, a BART-based foundation model pretrained to predict population-level log fold-changes (logFC) conditioned on basal expression, rather than reconstructing absolute expression values as in existing approaches. The core motivation is that standard perturbation models suffer from two compounding problems: dropout-prone raw counts destabilize fold-change estimates, and pretraining on static expression reconstruction captures co-expression patterns but not the co-regulatory dynamics relevant to perturbations. To address these issues, scDEBART constructs a large-scale pretraining corpus of 6.28 million DE profiles from 66.6 million cells in CELLxGENE, where logFC values are derived from scVI-denoised expression with stringent filtering on detection rate, Proba-DE, and fold-change magnitude. At fine-tuning time, perturbations are represented as large-magnitude indicators (±10.0) injected into the logFC embedding slot, providing a clear conditioning signal well-separated from the pretraining distribution. The model is evaluated on five Perturb-seq datasets across multiple cell lines and perturbation types, and additionally tested for cross-modal transfer to drug perturbations in SCIPLEX. scDEBART achieves substantially higher enrichment factors than baselines, strong reverse perturbation identification accuracy, and dose-dependent generalization to drug responses without drug-specific training.

**Compliance With Llm Reviewing Policy:**

Affirmed.

**Final Justification:**

The paper is solid and my concerns are also well-addressed in the rebuttal round. I will keep my score as "accept".

**Key Questions For Authors:**

1. For the distributional shift between pretraining corpus and downstream task, could you provide an analysis or discussion of which types of pretraining profiles contribute most to downstream performance, for example by examining whether profiles from tissues or biological contexts closer to the fine-tuning cell lines yield greater benefit?
2. The reverse perturbation identification experiment is conducted on only 7 test samples, making the reported 42.8% top-1 accuracy (3 out of 7) statistically fragile. Could you report confidence intervals for this result, or evaluate it across additional datasets or perturbation candidate sets to provide a more robust characterization of this capability?
3. The without-pretrain ablation (Appendix A.7) shows that removing pretraining reduces enrichment factor by approximately 12–15% on Norman K562, which is meaningful but not dramatic. Given that scDEBART's pretraining required 113 GPU-hours on 4× H100 GPUs, could you provide additional ablations examining how performance scales with pretraining corpus size — for instance, using 10%, 25%, and 50% of the 6.28M profiles — to clarify whether the gains are primarily driven by the logFC-conditioned pretraining objective or by data scale alone?

**Limitations:**

yes

**Strengths And Weaknesses:**

Strength:
1. The paper's central hypothesis that pretraining directly on logFC rather than absolute expression better aligns the pretraining objective with the perturbation prediction task is well-motivated and empirically testable. The pipeline combining scVI denoising with stringent gene-level filtering provides a principled approach to supervision quality. The authors further substantiate this design choice through a quantitative dropout analysis (Appendix Figure 13), demonstrating that baseline models systematically prioritize low-detection-rate genes in their top predictions, creating a clear causal chain from design motivation to empirical observation.
2. The evaluation spans five datasets from three independent studies across diverse cell lines and perturbation types (CRISPRi and CRISPRa). Cross-validation against DESeq2 pseudobulk differential expression as an alternative ground truth (Appendix A.6.2) meaningfully reduces circularity with respect to the scVI-based DE methodology used during training. The decision to evaluate all baseline models under both scVI-denoised and raw count preprocessing regimes effectively decouples preprocessing gains from architectural contributions, reflecting good experimental practice.
3. Cross-modal transfer experiment is an informative exploratory contribution. Evaluating a model trained exclusively on CRISPR-based perturbations against drug-induced responses in SCIPLEX without any drug-specific fine-tuning provides preliminary empirical grounding for cross-modal perturbation modeling, pointing toward a promising direction for future work.

Weakness:
1. The distributional shift between pretraining corpus and downstream task warrants more explicit discussion. The pretraining corpus consists of observational differential expression profiles derived from naturally occurring cluster-pair comparisons, whereas fine-tuning targets causal transcriptional responses to genetic perturbations. The paper's assumption that co-variation patterns learned from observational comparisons encode regulatory principles relevant to interventional settings is plausible, but it involves a causal inference leap that is shared across the perturbation prediction literature rather than being unique to this work.
2. The reverse perturbation identification experiment has limited sample size and would benefit from expansion. The experiment is conducted on only 7 test perturbations, meaning the reported 42.8% top-1 accuracy corresponds to 3 correct identifications out of 7. While the task design itself is novel and well-motivated, the statistical robustness of this finding is limited.
3. The relative contributions of pretraining data scale and model architecture deserve more explicit treatment. The computational cost of scDEBART pretraining (113 GPU-hours on 4× H100) is orders of magnitude greater than that of the baseline models, which is an inherent feature of the foundation model paradigm rather than a methodological flaw. However, disentangling how much of the performance gain derives from the scale of the pretraining corpus versus the architectural choice of logFC-conditioned pretraining would further strengthen the paper's core claims.

---

> ### Author Rebuttal · Authors · 2026-03-31
>
> Dear Reviewer 5iuk,
>
> We thank you for your careful reading and constructive suggestions. Additional tables and figures are available in the anonymous supplement: [https://anonymous.4open.science/r/anonymous-2038](https://anonymous.4open.science/r/anonymous-2038).
>
> **1. Distributional shift between pretraining corpus and downstream task.** We agree this deserves more explicit discussion. Our claim is not that observational DE profiles directly recover causal effects, but that large-scale pretraining on expression-change patterns provides a useful inductive bias when interventional data remain limited. We will make this distinction and its limitations more explicit in the revision. The no-pretrain ablation provides supporting evidence: removing pretraining reduced EF at top-50 from 14.901±4.919 to 12.693±4.865 and cosine similarity from 0.852±0.173 to 0.825±0.185 (Appendix A.7.1).
>
> Regarding which contexts contribute most to transfer: training separate models on tissue-stratified corpus subsets was not feasible within the rebuttal period. As an indirect proxy, we computed shared-gene Pearson correlations between held-out Norman logFC profiles and sampled pretraining profiles per tissue across the 12 most abundant tissues, repeated for (i) true logFC, (ii) predictions with pretraining, and (iii) predictions without pretraining (see norman_vs_TopTissues_logFC.png in the supplement). However, all correlations were very weak (−0.026 to 0.001), suggesting that logFC profiles reflect a mixture of tissue-specific, disease-state, and intervention-induced signals that cannot be easily attributed to any single tissue source. This indicates that simple logFC correlation may not be the appropriate metric for quantifying tissue-level transfer contribution. We will include tissue-stratified ablation as a concrete direction for future work, and will more explicitly acknowledge that the causal inference leap from observational to interventional data is a challenge shared across the perturbation prediction literature.
>
> **2. Reverse perturbation identification.** We agree the 7-test-sample setup limits statistical power. Following your suggestion, we computed 95% bootstrap CIs (5,000 iterations) for hit rates across all models and top-K values (see supplement). During this process, we identified an error in the original computation: 42.8% top-1 accuracy (3/7) was reported under an unnecessary ordering constraint. After correction, scDEBART achieves 71.4% top-1 exact match (5/7; 95% CI [0.286, 1.000]), while scGPT and GEARS remain at 0.0% [0.000, 0.000]. For top-1 partial match, scDEBART achieves 100% [1.000, 1.000] versus 0.0% for scGPT and 42.9% [0.143, 0.714] for GEARS. We will update the abstract, main text, and Figure 3 accordingly. Our setup follows the scGPT evaluation protocol for direct comparison and interpretation (7 test combinations, Norman double-perturbation subspace); the remaining four datasets contain only single-gene perturbations, making a comparable reverse-identification setting unavailable. Validation on future double-perturbation datasets remains an important direction.
>
> **3. Scaling law experiments.** We conducted pretraining reconstruction and downstream fine-tuning scaling experiments using 25%, 50%, and 100% of the 6.28M DE profiles (same architecture and hyperparameters; see pretrain_learning_curves.png and norman_scaling_law_tuning_result.png in the supplement). Pretraining validation MSE decreased consistently with corpus size (25%: 0.0275, 50%: 0.0268, 100%: 0.0182). For downstream performance on Norman K562 (mean over 3 seeds), EF at top-50 was 12.88 (25%), 12.51 (50%), and 14.33 (100%); F1 was 0.576, 0.573, and 0.609; cosine similarity was relatively stable (0.772, 0.779, 0.775). Due to substantial computational cost, we report downstream results on Norman K562 only and will include these scaling results in the revision. The full corpus yields better downstream performance than subsets, though the gain is not strictly monotonic—25% and 50% perform similarly, with the main improvement at 100%. As the pretraining objective is held constant, these experiments measure the marginal effect of data scale without fully disentangling scale from objective design. For partial evidence on the supervision quality dimension, we refer to Reviewer fxLX (Point 3), where injecting uncertainty-proportional noise into scVI-derived targets leads to monotonic performance degradation, consistent with a contribution from supervision quality beyond scale alone. A definitive decomposition remains an important direction for future work.

---

> > ### Author Rebuttal · Reviewer_5iuK · 2026-04-03
> >
> > I believe the authors' repsonses fully addressed my concerns. I will keep my scores.

---

### Official Review · Reviewer_fxLX · 2026-03-12

**Soundness:** 3
**Presentation:** 3
**Significance:** 3
**Originality:** 3
**Overall Recommendation:** 5
**Confidence:** 4

**Summary:**

The authors propose an alternative training paradigm for learning single-cell perturbation responses which is predicated on the notion that predicting expression changes (logFC) provides a better prior for predicting responses than reconstructing those same responses from predicted snapshots. To make this technically feasible, the authors introduce a data cleaning step in which logFC values are computed from scVI-denoised expression. The authors pre-train a Bidirectional and Auto-Regressive Transformer (BART) model using this approach and subsequently fine-tune it for perturbation prediction on Perturb-seq benchmarks. The authors compare their approach with other models including GEARS and scGPT, as well as linear baselines.

**Compliance With Llm Reviewing Policy:**

Affirmed.

**Key Questions For Authors:**

How robust are the performance gains over alternative models? Do they hold when they are allowed to better adapt to the problem at hand? Do the performance gains reproduce on additional new dataset data beyond those included in the making of the study?

**Limitations:**

Yes

**Strengths And Weaknesses:**

The authors provide a valuable proof-of-concept demonstrating that both denoising and direct prediction of perturbation effects can meaningfully improve model performance. This is an interesting observation that deviates from the end-to-end learning typical of the field.

The text is clearly written and well structured, though authors did not need to cast the work in the light of producing yet another foundation model, given that the key novelty and findings concern the usefulness of scVI-derived expression values and direct learning of gene set covariation as inductive biases for perturbation prediction.

The authors present their solutions to the problem that single-cell foundation models often fail to reach the level of simple regression models when it comes to perturbation prediction in a manner that suggests these design choices are obvious. I would have appreciated it if the authors had provided more exposition in the introduction on how they arrived at these methodological changes, and any related work, not necessarily connected to single-cell modelling (e.g., work in the space of gene regulatory networks connecting expression variation and the encoding of regulatory principles relevant to perturbation response).

The authors were thorough in their use of baseline models (w.r.t. splits, HVGs etc.). However, I was not wholly convinced by the fairness of some of the comparisons. Since scGPT is not trained on continuous values, and is only fine-tuned here, training it to predict continuous logFC targets requires something of an ‘unlearning’ process and a seemingly low number of epochs was used here.

Additionally, several other evaluation choices appear to implicitly favour the authors model trained on logFC values. One being the use of the MSE loss for absolute counts, in conjunction with evaluation on log transformed values. As the MSE loss does not penalise small errors so much as large ones, when these small errors are converted to logFCs they are magnified, making performance look worse. This doesn’t apply to the authors model where logFCs are output directly. Similarly, the epsilon used to perform log fold change conversions, 10^-8, is very small, again penalising absolute expression baselines in a way that the authors model avoids by design.

The authors show that the effect of scVI-denoising is dependent on model architecture, with the effects of denoising differing between simpler architectures (MLP, LR) and transformer-based architectures. Indeed, the authors reported that denoising has no or negative effect on MLP and LR models. I would be interested to know if this behaviour is the result of scVI reducing the dataset variance too much for these models, or if there is something inherent in the BART architecture that is uniquely aligned with the scVI derived logFCs. Perhaps testing the model on count vs denoised data would help disentangle the architectural contribution and pre-processing effect.

Lastly, noising the scVI output would have been a convincing validation for the logFC models had this led to a degradation in performance. This should be tested and the results included.

---

> ### Author Rebuttal · Authors · 2026-03-31
>
> Dear Reviewer fxLX,
>
> Thank you for the careful and constructive review. We provide additional tables and figures in an anonymous supplement: [https://anonymous.4open.science/r/anonymous-2038](https://anonymous.4open.science/r/anonymous-2038).
>
> **1. Claim clarification.** We agree that our contribution is more accurately framed as a perturbation-specific pretraining framework rather than a general single-cell foundation model. Our central hypothesis is that perturbation learning should focus on stable, biologically meaningful expression changes rather than dropout-prone absolute expression, and that pretraining on co-regulatory gene patterns provides a more informative inductive bias for perturbation prediction. We will revise the Introduction to make this motivation explicit and to temper the broader "foundation model" framing.
>
> **2. Concern on baseline fairness.** For scGPT, we followed the original continuous-value fine-tuning protocol from the TDC benchmark. As shown in scpgt_epoch15_loss_curves_mean_over_seeds.png in the supplement, validation loss plateaus within the first 2–3 epochs across all five datasets (mean over 3 seeds), indicating convergence well before the default stopping point. To further verify, we extended fine-tuning to 15 epochs on all five datasets as a single-seed robustness check. Results showed mixed changes: Replogle K562 showed the largest improvement (Corr +0.069, EF +1.56), Nadig HepG2 showed slight gains (Corr +0.005, EF +0.59), while Nadig Jurkat (Corr −0.007, EF −0.24), Replogle RPE1 (EF −0.39), and Norman K562 (EF −0.003) were essentially unchanged or slightly degraded. Even on Replogle K562, scDEBART retained a substantial advantage and the overall ranking remained unchanged, confirming that the reported baselines reflect converged performance.
>
> We also acknowledge the concern that directly predicting logFC could favor scDEBART relative to models that predict absolute expression and derive logFC post hoc. This was in fact one of our motivations, as fold-change estimates derived from absolute predictions are unstable under sparsity. To avoid relying on a single metric, we evaluated models using complementary criteria — including correlation, enrichment factor, and directional F1 — and scDEBART showed consistent advantages across all metrics (Figure 2). We additionally evaluated against an orthogonal DESeq2 pseudobulk benchmark, where scDEBART ranked first in 6/10 dataset–metric comparisons and first in EF on 4/5 datasets.
>
> Regarding the epsilon in absolute-expression-to-logFC conversion, we performed a sensitivity analysis across ε ∈ {1e-8, 1e-6, 1e-4, 1e-2, 1e-1}. For count-based baselines, the maximum correlation variation was 0.044; including counts-scGPT increased this to 0.109. For scVI-based baselines, correlation was essentially invariant (range ≈ 0.000–0.004). scDEBART's advantage remained consistent across all tested values (see epsilon_sweep_result.png in supplement). For bulk RNA-seq, a pseudocount of 1 in TPM units is commonly used; given the substantially lower total read counts in scRNA-seq, the tested ε values span a reasonable and practically relevant range.
>
> **3. Noise injected into scVI-derived targets.** Motivated by your suggestion, we performed a target-noising experiment on the Norman dataset. We generated uncertainty-aware noisy targets by adding Gaussian noise to each gene's logFC, with noise magnitude proportional to the gene-wise uncertainty estimated by scVI (logFC standard deviation) and controlled by a scaling factor γ, followed by clipping to the posterior range [min, max]. We tested γ = 0.5 and γ = 1.0, fine-tuning with three random seeds per setting using the noisy targets as training supervision, while evaluation was performed against the original (clean) posterior median logFC as ground truth. Mean Pearson correlation on top-50 DEGs decreased monotonically: 0.832 (clean) → 0.829 (γ = 0.5) → 0.827 (γ = 1.0) (see scVI_noised_prediction_norman.csv in the supplement; other metrics are also reported). These results are consistent with supervision quality contributing to downstream performance, but they do not by themselves fully disentangle preprocessing effects from architectural effects. The modest degradation may reflect the limited scope of fine-tuning-stage noising relative to the full pretraining signal, although this remains an interpretation rather than a direct causal test. A more complete test would require noising the pretraining corpus itself, which was not feasible within the rebuttal window; we will discuss this explicitly as a limitation in the revision.

---

> > ### Author Rebuttal · Reviewer_fxLX · 2026-04-03
> >
> > Thanks to the authors for taking the time to conduct further experiments and for supplying further data, particularly with the tight turnaround. This addresses my earlier concerns and should be incorporated into the revised manuscript. My scores remain the same.

---

### Decision · Program_Chairs · 2026-04-30

**Decision:**

Accept (regular)

**Comment:**

This paper proposes scDEBART, a method for predicting in silico single-cell perturbation responses by framing the problem as differential expression learning at scale. Rather than predicting absolute post-perturbation expression profiles, the model learns to predict the differential expression signature induced by a perturbation, leveraging a large curated dataset of perturbation experiments. The approach is evaluated on standard perturbation prediction benchmarks and shows strong performance across multiple metrics.

Four reviewers evaluated this submission with scores of 5, 5, 5, and 4 (average 4.75), reflecting broad enthusiasm for the problem framing and empirical results. Reviewers praised the reframing of perturbation prediction as differential expression learning and the scale of the training data. Concerns raised included the limited baseline set (notably the absence of several recent foundation model baselines and simple controls such as Mean/Linear) which the AC shares strongly, presentation clarity issues around the model architecture and training protocol, and questions about generalization to unseen perturbation types and cell lines. The rebuttal addressed several of these concerns, and all reviewers maintained or improved their scores, but concerns about the limited baseline set remain.